

# Summer ozone in the Northern Front Range Metropolitan Area: Weekend-weekday effects, temperature dependences and the impact of drought

Andrew A. Abeleira[1], Delphine K. Farmer[1]

1. Department of Chemistry, Colorado State University, Fort Collins, CO, 80523, USA

*Correspondence to*: Delphine K. Farmer (delphine.farmer@colostate.edu)

**Abstract.** Contrary to most regions in the U.S., ozone in the Northern Front Range Metropolitan Area (NFRMA) of
Colorado was either stagnant or increasing between 2000 and 2015, despite substantial reductions in $NO_x$ emissions.
We used available long-term ozone and $NO_x$ data in the NFRMA to investigate these trends. Ozone increased from
weekdays to weekends for a number of sites in the NFRMA with weekend reductions in $NO_2$ at two sites in downtown
Denver, indicating that the region was in a $NO_x$-saturated ozone production regime. The stagnation and increases in
ozone in the NFRMA are likely the result of (1) decreasing $NO_x$ emissions in a $NO_x$-saturated environment, and (2)
increased anthropogenic VOC emissions in the NFRMA. Further investigation of the weekday-weekend effect showed
that the region outside of the most heavily trafficked Denver area was transitioning to peak ozone production towards
$NO_x$-limited chemistry. This transition implies that continued $NO_x$ decreases will result in ozone being less sensitive
to changes in either anthropogenic or biogenic VOC reactivity in the NFRMA. Biogenic VOCs are unlikely to have
increased in the NFRMA between 2000 and 2015, but are temperature dependent and likely vary by drought year.
Ozone in the NFRMA has a temperature dependence, consistent with biogenic VOC contributions to ozone production
in the region. We show that while ozone increased with temperature in the NFRMA, which is consistent with a $NO_x$-
saturated regime, this relationship is suppressed in drought years. We attribute this drought year suppression to
decreased biogenic isoprene emissions due to long-term drought stress.
**1. Introduction**
Tropospheric ozone ($O_3$) is detrimental to human health, impacting asthma attacks, cardiovascular disease, missed
school days, and premature deaths. Based on these impacts, the Environmental Protection Agency (EPA) projects that
reducing the $O_3$ standard to the new 70 $ppb_v$ 8-hour average will result in health benefits of $6.4-13 billion/yr (EPA,
2014). $O_3$ also damages plants, reducing agricultural yields (Tai et al., 2014). Using crop yields and ambient $O_3$
concentrations for 2000, Avnery et al. (2011) estimate the loss of $11-18 billion/yr worldwide as a result of the
reduction of staple worldwide crops (soybean, maize, and wheat) from $O_3$ damage. During summer months, the
Northern Front Range Metropolitan Area (NFRMA) of Colorado consistently violated the pre-2016 U.S. EPA
National Ambient Air Quality Standard (NAAQS) of 75 $ppb_v$ fourth-highest daily maximum 8-hour average (MDA8)
ambient $O_3$ concentration, despite proposed reductions in anthropogenic emissions (CDPHE, 2014). The NFRMA has
been an $O_3$ non-attainment zone since 2008 (CDPHE, 2009), prompting the Colorado Air Pollution Control Division
and the Regional Air Quality Council to develop the Colorado Ozone Action Plan in 2008 to target key $O_3$ precursors:
volatile organic compounds (VOCs) and $NO_x$ ($NO+NO_2$)(CDPHE, 2008). Despite these control efforts, 2013 was the
NFRMA's fourth year in a row to exceed the federal $O_3$ standard (CDPHE, 2016), and the eight NFRMA non-
attainment counties, with their combined population >3.5 million, exceeded the MDA8 75 $ppb_v$ $O_3$ standard 9-48 days
between 2010 and 2012 (AMA, 2015). However, Colorado must comply with the new 70 $ppb_v$ MDA8 standard by
2018. In order to accurately design and implement $O_3$ reduction schemes, a thorough understanding of local $O_3$ trends
and chemistry is required.

34          Ground-level or boundary layer $O_3$ depends on local production, transport, and meteorological parameters:

$$\frac{\partial [O_3]}{\partial t} = P(O_3) + \frac{w_e O_3 - u_d [O_3]}{H} - \nabla \times (v[O_3]) \qquad (1)$$
where $\partial [O_3] / \partial t$ represents the time rate of change of $O_3$ concentration, $P(O_3)$ is the instantaneous net photochemical
$O_3$ production rate (production – loss), $w_e O_3 - u_d [O_3] / H$ represents the entrainment rate ($w_e$) of $O_3$ in and deposition





rate ($u_d$) of $O_3$ out of the mixing layer height (H), and $\nabla \times (v[O_3])$ describes the advection of $O_3$ mixing layer height.
Briefly, ground-level $O_3$ is driven by a catalytic chain that is initiated by $RO_2$ production from VOC oxidation (R1),
and propagated by local $NO_x$ emissions (R2,3).

$$RH + OH + O_2 \rightarrow RO_2 + H_2O \qquad (R1)$$

Chain propagation occurs through reactions between $HO_2$ or $RO_2$ radicals with NO to form $NO_2$ (R2a,b, R3), which
is photolyzed (R4) and leads to net $O_3$ formation (R5). Reactions between NO and $O_3$ also produces $NO_2$ (R6),
leading to a null cycle with no net $O_3$ production. Alkoxy (RO) radicals form carbonyl-containing compounds and
$HO_2$ (R7).

$$RO_2 + NO \rightarrow RO + NO_2 \qquad (R2a)$$

$$RO_2 + NO \rightarrow RONO_2 \qquad (R2b)$$

$$HO_2 + NO \rightarrow NO_2 + OH \qquad (R3)$$

$$NO_2 + hv \rightarrow NO + O(^3P) \qquad (R4)$$

$$O(^3P) + O_2 \rightarrow O_3 \qquad (R5)$$

$$NO + O_3 \rightarrow NO_2 + O_2 \qquad (R6)$$

$$RO + O_2 \rightarrow R'CHO + HO_2 \qquad (R7)$$

For every VOC that enters the cycle, approximately two $NO_2$ radicals are produced – but the resulting carbonyl-
containing compounds and organic nitrates can be repeatedly oxidized or photolyzed, further propagating the $P(O_3)$
chain. Chain termination occurs through $RO_2$ and $HO_2$ self-reactions to form peroxides (dominant termination
reactions in the "$NO_x$-limited regime"), OH and $NO_2$ reactions to form $HNO_3$ ("$NO_x$-saturated"or "VOC-limited"
regime), or $RO_2$ and $NO_x$ reactions to form organic nitrates ($RONO_2$) or peroxyacyl nitrates ($RC(O)O_2NO_2$).
Formation of organic and peroxyacyl nitrates suppresses $P(O_3)$, but does not shift the cross-over point between $NO_x$-
limited and $NO_x$-saturated $P(O_3)$ regimes (Farmer et al., 2011). This cross-over point of maximum, or peak, $O_3$
production is controlled by the chain termination reactions, and is sensitive to the $HO_x$ production rate and thus VOC
reactivity. Decreasing $NO_x$ is an effective $O_3$ control strategy in a $NO_x$-limited system, but will increase $O_3$ in a $NO_x$-
saturated system. Controls for $NO_x$-saturated systems often focus on reducing anthropogenic VOC reactivity, and/or
suppressing $NO_x$ emissions sufficiently that the system becomes $NO_x$-limited.
Trends in $O_3$ for 2000 – 2015 varied across the United States (EPA, 2016a). Using the annual 4[th] maximum of daily
8-hour averages (MDA-8), the EPA reported a 17% decrease in the aggregated national average $O_3$. However, regional
trends deviated substantially from the national average. For example, the EPA reported a 25% decrease in $O_3$
throughout the southeast, while the northeast shows a 16% decrease. Smaller decreases in $O_3$ occurred in the northern
Rockies (1%), the southwest (10%) and the west coast (4-10%). These $O_3$ reductions are concurrent with national
reductions in $O_3$ precursors of 54% for $NO_x$, 21 % for VOCs, and 50% for CO (EPA, 2016b). Due to the non-linear
behavior of $O_3$ chemistry described above, reductions in $O_3$ precursors do not necessarily result in reductions of
ambient $O_3$. Cooper et al. (2012) reported that 83%, 66%, and 20% of rural eastern U.S. sites exhibited statistically
significant decreases in summer $O_3$ at the 95[th], 50[th], and 5[th] percentiles (1990-2010). No increases in $O_3$ occurred at
any sites, indicating that local emission reductions have been effective in those regions. In contrast, $O_3$ in the western
US followed a very different trend: only 8% of western U.S. sites exhibited decreased $O_3$ at the 50[th] percentile; the 5[th]
percentiles for $O_3$ at 33% of the sites actually increased. These increases were larger for the lower percentiles,
indicating that while local emissions reductions may have been effective at some sites, increased background $O_3$ offset
the improvement.
Lefohn et al. (2010) found that $O_3$ decreased across many U.S. sites at a less rapid pace during 1994-2008 than during
1980-2008, indicating that $O_3$ improvements had leveled off by the late 2000s. The leveling off could be a result of
either slowed precursor emissions reductions, which is contrary to the EPA estimates, or, more likely, shifting $O_3$
chemistry regimes as precursor emissions are changing. Lefohn et al. (2010) reported that the distributions of high





and low hourly O$_3$ values narrowed toward mid-level values in the 12 cities studied, consistent with a reduction in
domestic O$_3$ precursors and possibly increased transport of O$_3$ precursors from east Asia. A number of modeling and
measurement studies have also reported increased baseline O$_3$ in the western U.S. due to the transport of O$_3$ precursors
from east Asia (Cooper et al., 2010;Parrish et al., 2004;Pfister et al., 2011;Weiss-Penzias et al., 2006). These studies
questioned the effectiveness of local precursor emission reductions in controlling local O$_3$ in impacted regions.
Cooper et al. (2012) showed that the intermountain West is an intriguing environment with potentially increasing
background O$_3$. The NFRMA is of particular interest due to the challenge in effective O$_3$ regulation, its growing
population and the dominantly anthropogenic sources of O$_3$ precursors. VOCs have been well-studied in the region,
with a particular focus on the Boulder Atmospheric Observatory (BAO) in Erie, CO (e.g. Gilman et al., 2013;McDuffie
et al., 2016;Pétron et al., 2012;Swarthout et al., 2013;Thompson et al., 2014). VOC composition in the NFRMA was
heavily influenced by oil and natural gas (ONG) sources, as well as traffic. In winter 2011, ~50% of VOC reactivity
was attributed to ONG-related VOCs and ~10% to traffic (Gilman et al., 2013;Swarthout et al., 2013). Recent studies
have shown that ONG and traffic contributed up to 66% and 13% of the VOC reactivity respectively at BAO in
mornings for both spring and summer 2015, but that biogenic isoprene was a large, temperature-dependent component
of VOC reactivity in the summer, contributing up to 49% of calculated daytime VOC reactivity (Abeleira et al., 2017).
We note that the anthropogenic VOCs were typically lower in 2015 than previous measurements, pointing to the
complex roles of meteorology, transport and local emissions. In contrast, observed isoprene in summer 2012 was
much lower than summer 2015, likely due to shifting drought conditions. While temperatures across the two summers
were similar, 2012 was a widespread drought year in the region, and 2015 was not; drought is typically associated
with suppressed biogenic VOC emissions. Local anthropogenic and biogenic sources are not the only VOC sources
in the region: longer-lived VOCs consistent with transport have also been observed (21-44% of afternoon reactivity
in 2015), and smoke from both local and long-distance wildfires impacted air quality in the NFRMA in punctuated
events. This smoke was sometimes, but not always, associated with elevated O$_3$ (Lindas et al., 2017).
The impact of a changing climate on air quality is poorly understood due to the complex climate-chemistry interactions
and numerous feedbacks (Jacob and Winner, 2009;Palut and Canziani, 2007). However, increasing temperature is
expected to increase O$_3$ (Bloomer et al., 2009;Jacob and Winner, 2009;Palut and Canziani, 2007). The O$_3$-temperature
relationship is attributed to (1) temperature-dependent biogenic VOC emissions that provide a source of VOCs for
OH oxidation leading to increased HO$_x$ cycling (Guenther, 2006;Guenther et al., 1996), (2) thermal decomposition of
peroxyacetylnitrate (PAN) to HO$_x$ and NO$_x$ (Fischer et al., 2014;Singh and Hanst, 1981), and (3) increased likelihood
of favorable meteorological conditions for ozone formation (*i.e.* high insolation, stagnation, circulating wind patterns)
(Reddy and Pfister, 2016;Thompson et al., 2001). In addition, increased temperatures and changing soil moisture could
alter soil emissions of NO$_x$. Due to the non-linearity of P(O$_3$) chemistry as a function of NO$_x$, the increased VOC and
NO$_x$ emissions associated with warming can either increase or decrease P(O$_3$) depending on local NO$_x$ concentrations
(i.e. NO$_x$-limited vs. NO$_x$-saturated). Interactions between climate change and regional-scale meteorology are
complex, and may also impact O$_3$. High and low O$_3$ in the U.S is coupled to a variety of meteorological parameters
including planetary boundary layer (PBL) heights (White et al., 2007;Reddy and Pfister, 2016), surface temperatures
(Bloomer et al., 2009), soil-moisture and regional winds (Davis et al., 2011;Thompson et al., 2001). PBL height is
coupled to increased temperatures, reduced cloud cover, stronger insolation, and lighter circulating wind patterns with
higher 500 hPa heights correlating to higher average July O$_3$ in the NFRMA (Reddy and Pfister, 2016).
In this paper, we used temperature, O$_3$, and NO$_2$ data from 2000-2015 at multiple sites in the NFRMA to investigate
why O$_3$ has not decreased in the region despite decreases in NO$_x$. We used a weekday-weekend analysis to elucidate
the NO$_x$ regime for P(O$_3$) in Denver, and explored the temperature dependence of O$_3$ and the role of drought in
influencing that relationship in the NFRMA.
**2. Methods**
**2.1 Measurement sites**
We used publicly available O$_3$, NO$_2$ and temperature data (https://aqs.epa.gov/aqsweb/documents/
data_mart_welcome.html) from eight sites in the NFRMA (Fig. 1, Table 1). The CAMP site is 1 mile east of the I-25
interstate highway in downtown Denver. O$_3$ data was available for 2005 – 2007 and 2012 – 2015, while NO$_2$ data was



available for 2001 – 2007 and 2010 – 2015. Welby is roughly 8 miles northeast from the CAMP site, and is adjacent
to a large lake and less than 1-mile west of the Rocky Mountain Arsenal open space. $O_3$ data was available for 2000 –
2009 and 2011 – 2015, while $NO_2$ data was available for 2001 – 2002, 2004 – 2005, 2007 – 2008, and 2010 – 2015.
The Carriage site is <1 mile west of the I-25 interstate at the same latitude as the CAMP site. $O_3$ data was available
for 2000 – 2012 for the Carriage site. The Fort Collins site is adjacent to Colorado State University near downtown
Fort Collins. $O_3$ data was available for 2000 – 2015. The Greeley site was located on the southeast side of Greeley and
<1 mile south of CO state highway 34. $O_3$ data was available for 2002 – 2015. The Rocky Flats site is in a rural area
adjacent to the Rocky Flats Wildlife Refuge <15 miles south of Boulder. The I-25 site is adjacent to the I-25 interstate
2-miles south of the Carriage and CAMP sites, and intercepts fresh $NO_x$ emissions directly from the I-25 interstate.
$NO_2$ data was available for 2015, but not $O_3$. The La Casa site is <1 mile west of the I-70 and I-25 interstate junction.
$O_3$ and $NO_2$ data were available for 2015. Temperature data was available for all sites for all years.
**2.2 Ozone and $NO_2$ data treatment**
Ambient $NO_2$ concentrations were measured by chemiluminescence monitors equipped with molybdenum oxide
converters. These monitors are used as the EPA Federal Reference Method for monitoring ambient $NO_2$
concentrations, and have a known interference from nitric acid and organic nitrates (Dunlea et al., 2007). The true
ambient $NO_2$ mixing ratio is a component of the reported values. $NO_2^*$ will be used in this manuscript to refer to the
EPA $NO_2$ measurements, which includes the interference, and can be considered to be a proxy for total reactive
nitrogen oxides ($NO_y$). While the absolute $NO_2^*$ concentration will be greater than $NO_2$ but less than $NO_y$, trends in
$NO_2^*$ provided insight on trends in local $NO_x$ emissions. The $O_3$ and $NO_2^*$ mixing ratios are filtered to summer months
(June 1 – August 31), and averaged to a daytime value (10:00 am – 4:00 pm local). A site was excluded for a given
year when <50% of data is available for that summer.
**2.3 Trend analysis**
Following the analyses of Cooper et al. (2012), the statistical significance of the linear trends were tested with a
standard F-test with the null hypothesis that there is no linear trend ($R^2 = 0$). The null hypothesis was rejected with a
confidence level ≥ 95% if the probability (p) associated with the F-statistics was low ($p \leq 0.05$).
**3 Results and Discussion**
**3.1 Long term trends in $O_3$ and $NO_2^*$ in the Northern Front Range Metropolitan Area**
Contrary to most other places in the U.S., $O_3$ in the NFRMA was either stagnant or increasing between 2000 and 2015,
despite substantial decreases in $NO_x$ emissions. At most sites in the eastern U.S. and some on the west coast, $O_3$ was
decreasing at all percentiles. In the NFRMA, however, five out of six monitoring sites exhibited no change or
increasing $O_3$ at the 50th and 95th percentiles in the 2000 – 2015 period (Fig. 2). The 5th percentile is often taken as
background $O_3$. With the exception of the Greeley site, the 5th percentile of $O_3$ increased across the NFRMA between
2000 and 2015. However, only the downtown Denver CAMP site had statistically significant increases in $O_3$ of 2.6 ±
0.9, 2.3 ± 0.3, and 1.8 ± 0.7 ppb$_v$/yr for the 5th, 50th, and 95th percentiles, respectively. The Welby site had increases
of 1.5 ± 0.5, 1.3 ± 0.4, and 1.4 ± 0.4 ppb$_v$/yr from 2000 – 2015 (Fig. 2b), but with a statistical significance at only the
95th percentile. Cooper et al. (2012) reported that the Welby site exhibited no statistically significant increase in $O_3$
from 1990 – 2010, contrary to what we found for 2000 – 2015 at the 95th percentile.
The increasing $O_3$ trends in the NFRMA occurred despite reductions in $NO_x$. $NO_2^*$ at the CAMP site decreased
significantly from 2000 at a rate of 1.2 ± 0.2 and 1.5 ± 0.2 ppb$_v$/yr for the 50th and 95th percentiles for CAMP (Fig. 3).
Welby exhibited a non-significant decreasing $NO_2^*$ trend at the 95th percentile of 0.5 ± 0.3 ppb$_v$/yr (Fig. 3). The
increased $O_3$ may be due to increased summer temperatures in Colorado, increased regional baseline $O_3$, or increased
local $P(O_3)$ from unknown emission sources (Cooper et al., 2012). VOC emissions steadily increased in Colorado
from 2000 to 2012 according to the EPA NEI-2014. To the best of our knowledge, the NFRMA does not have any
long-term VOC datasets, but the EPA NEI-2014 for Colorado provided an estimate for yearly anthropogenic VOC
(AVOC) emissions (EPA, 2016b). All categories of AVOC emissions decreased slightly from 2000 – 2015, except
for petroleum related VOCs which increased from 7.4 x $10^3$ tons in 2000 to 2.6 x $10^5$ tons in 2011 with a decrease to
1.5 x $10^5$ tons in 2015 (Fig. 4). However, we note the NEI is only an estimate and does not include biogenic sources





of VOCs, which can contribute substantially to VOC reactivity in the region, but vary substantially from year to year
(Abeleira et al., 2017). The increased $O_3$ is thus unsurprising for the 2000 – 2015 timeframe. The long-term reduction
in $NO_x$ with increasing VOC emissions concurrent with an increase in $O_3$ at both sites suggests that the downtown
Denver sites were in a $NO_x$-saturated $P(O_3)$ regime, and as $NO_2^*$ decreases and VOC reactivity increases, $P(O_3)$ was
increasing towards peak production.

**3.2 Weekday-Weekend effect in Denver, CO**

The 'weekday-weekend effect' describes how emissions can be statistically different on weekdays versus weekends,
resulting in different secondary chemistry. This effect can be used to elucidate information about local chemical
regimes (i.e. CARB, 2003;Murphy et al., 2007;Fujita et al., 2003;Warneke et al., 2013;Pollack et al., 2012;Cleveland
et al., 1974;Heuss et al., 2003). Traffic patterns in urban regions are different between weekdays and weekends, with
heavier traffic and thus higher $NO_x$ on weekdays due to rush-hour and commercial trucking patterns. VOCs are
expected to be stable across the week, as major VOC sources do not vary by day-of-week. Despite this drop in traffic,
$O_3$ can be higher on weekends than on weekdays if the system is in a $NO_x$-saturated regime because decreased $NO_x$
increases $P(O_3)$, while decreased NO also reduces $O_3$ titration to $NO_2$ (Fujita et al., 2003;Heuss et al., 2003;Marr and
Harley, 2002;Murphy et al., 2007;Pollack et al., 2012;Pusede and Cohen, 2012). Thus urban regions, which are often
$NO_x$-saturated, tend to follow a day-of-week pattern in both $NO_x$ and $O_3$ (Fujita et al., 2003;Heuss et al., 2003;Pusede
and Cohen, 2012), while rural and semi-urban areas often experience no change in $NO_x$ or $O_3$ from weekdays to
weekends. Rural regions have a lower population density, less defined daily traffic patterns, and minimal or no
commercial trucking (Heuss et al., 2003). The weekday-weekend effect typically relies on the assumption that the
VOC reactivity and thus $HO_x$ production is unchanged between the weekend and weekday. However, this is not always
the case, as decreased weekend $NO_x$ reduces $NO_x$+OH reactions, and thereby increases weekend OH and increased
$O_3$ (Warneke et al., 2013). Few studies of VOCs in the NFRMA exist, but our previous work found no significant
difference in measured VOC reactivity at the BAO site between weekends and weekdays in summer 2015 (Abeleira
et al., 2017).
In the NFRMA, long-term (i.e. 10+ years) $NO_2^*$ datasets only existed at the CAMP and Welby sites. Two sites in
Denver added $NO_2^*$ measurements in 2015, the I-25 and La Casa sites. The CAMP, I-25, and La Casa sites are all
located within a 4-mile radius that straddles the I-25 motorway; are surrounded by a dense network of roads,
businesses, and industrial operations; and experience high traffic density. Welby was located roughly 8-miles northeast
from the three other sites, and borders a large lake and the Rocky Mountain Arsenal open space. Welby was thus more
'suburban' than the other sites. Median $NO_2^*$ at CAMP has decreased from 37 $ppb_v$ in 2003 to 13 $ppb_v$ in 2015. The
median weekday I-25 and La Casa $NO_2^*$ mixing ratios in 2015 were similar to CAMP in 2007 (Fig. 5) indicating that
although $NO_2^*$ emission reductions have been effective in the region, mixing ratios in Denver are very site specific
An observable weekday-weekend effect in $NO_2^*$ existed for all sites with $NO_2^*$ data in all years except for Welby in
2007 (Fig. 5). $NO_2^*$ decreased by 20-50% from weekdays to weekends. Assuming that meteorology doesn't
systematically change between weekends and weekdays, we consider the weekend-weekday effect in $O_3$ to be
indicative of changes in $P(O_3)$ due to lower $NO_x$. Figure 6 follows the analysis of Pusede and Cohen (2012), presenting
summer average weekday and weekend $O_3$ values for Welby and CAMP with the values tethered for each year. The
values followed a curve similar to a modeled $P(O_3)$ curve, and indicates that reductions in $NO_x$ emissions from 2000
to 2015 have placed $O_3$ production in the Denver region in a transitional phase from $NO_x$-saturated to peak $P(O_3)$.
Regions that have higher $NO_x$ should observe greater impacts from changing VOCs than those that are closer to the
peak $P(O_3)$. This analysis also suggested that continued reduction of $NO_x$ would shift the system to a $NO_x$-limited
regime, in which changes in VOC reactivity due to shifting anthropogenic or biogenic emissions would have little
effect on $O_3$.
The average change in $O_3$ ($\Delta O_3$) and $NO_2^*$ ($\Delta NO_2^*$) from weekend to weekday is plotted as a function of year for the
two available NFRMA sites (Fig. 7a, Fig. 7b). A positive $\Delta O_3$ reflects a higher $O_3$ concentration on the weekend than
weekday, consistent with a $NO_x$-saturated system. The weekday-weekend effect decreased from 2000 to 2015 for five
of the six sites, all with similar $\Delta O_3$. This is consistent with the decreased regional $NO_x$ emissions, which would move
the system from $NO_x$-saturated to peak $P(O_3)$. The CAMP site was the exception, and consistently had a larger $\Delta O_3$
than the other sites. This was consistent with the CAMP site's higher $NO_2^*$ relative to Welby and the 30-50% decrease




in $NO_2$* from weekdays to weekend. Measured $NO_2$* decreased at both CAMP and Welby (Fig. 3), although the
$\Delta NO_2$* at CAMP and Welby was unchanged, with average $NO_2$* of -11 ± 3 $ppb_v$ and -1.7 ± 0.9 $ppb_v$, respectively.
Thus while absolute $NO_x$ emissions have changed, weekly traffic patterns have not. Applying a one-sided linear
regression to the five-site $\Delta O_3$ median for 2001-2015 yielded a statistically significant decreasing trend of -0.5 ± 0.1
$ppb_v$/yr with an $r^2$ = 0.55. The $\Delta O_3$ decreased across the NFRMA outside of the highest traffic regions in Denver, again
consistent with the hypothesis that the NFRMA $P(O_3)$ regime has transitioned from $NO_x$-saturated chemistry towards
peak $P(O_3)$. Two specific sites, Greeley and Rocky Flats, show negative $\Delta O_3$ values in recent years, suggesting that
those sites have, at least in those specific years, transitioned to $NO_x$-limited chemistry.
Collectively, this weekend-weekday analysis suggested that the region is NOx-saturated, but transitioning to a NOx-
limited region. Increases in $O_3$ are likely due to a combination of decreasing $NO_x$ and increasing VOC emissions.
While the lack of long-term VOC measurements prevents identification and quantification of those VOC sources, the
NEI suggested that petroleum-related VOCs have increased.
**3.3 The $O_3$-temperature penalty in the NFRMA**
Increasing temperature can increase $P(O_3)$ by enhancing biogenic and evaporative VOC emissions, but has variable
impacts on the weekday-weekend effect as a result of changing $NO_x$ emissions (Pusede et al., 2014). We showed that
while $O_3$ increased with temperature in the NFRMA, consistent with a $NO_x$-saturated regime, this relationship was
variable year to year. Ambient $O_3$ was correlated with increasing temperature across the U.S. (Bloomer et al.,
2009;Jacob and Winner, 2009;Pusede et al., 2014). While one study in the NFRMA from summer 2012 found that
biogenic VOCs (*i.e.* isoprene) had a minor impact on VOC reactivity (McDuffie et al., 2016), Abeleira et al. (2017)
found that isoprene contributed up to 47% of VOC reactivity on average in the late afternoon in summer 2015.
Studying the temperature dependence of $O_3$ allowed us to investigate the extent to which biogenic VOCs influenced
$P(O_3)$ in the NFRMA and the interannual variability in those temperature-dependent VOC sources, as well as the shift
from a $NO_x$-saturated to $NO_x$-limited $P(O_3)$ regime. $NO_x$-saturated regimes should be sensitive to changes in VOC
reactivity, while $NO_x$-limited systems should not. We note that while anthropogenic VOCs, such as solvents, may be
temperature dependent and contribute to this trend, we only observed temperature trends in isoprene at the BAO site
in 2015 – though we acknowledge that the observed VOC suite in that study was limited (Abeleira et al., 2017).
$O_3$ in the NFRMA demonstrated a clear temperature dependence at all percentiles for all sites, but with slopes that
vary by site and year (Fig. 8, Fig. 9). The NFRMA appears to be $NO_x$-saturated or near peak $P(O_3)$ for all years,
consistent with temperature dependent biogenic emissions having large impacts on ambient local $O_3$. The variance in
the $O_3$-temperature dependence was likely external to meteorological effects. High temperature and linked
meteorological parameters such as high 500 hPa heights, and stagnant winds, or circulating wind patterns do indeed
correlate with high $O_3$ events in Colorado (Reddy and Pfister, 2016), but those parameters should not affect the $O_3$
temperature relationship.
Figure 8a shows daytime, summer $O_3$ averaged in 3°C temperature bins for CAMP, Fort Collins, and Rocky Flats for
years in which data was available at all sites. For every temperature bin, $O_3$ was higher at Rocky Flats than at Fort
Collins, and both were higher than at CAMP. The Rocky Flats site was the most rural of the chosen sites adjacent to
the 4,000 acre Rocky Flats Wildlife Refuge, but was <15 miles from downtown Boulder. Rocky Flats likely had higher
$O_3$ because it was downwind of both $NO_x$ (Boulder, Denver) and VOC sources (forested regions in the neighboring
foothills), had fewer nearby fresh $NO_x$ sources and thus less $NO+O_3$ titration, and experienced enhanced $P(O_3)$ due to
the region being near at the cross-over point between $NO_x$-saturated and $NO_x$-limited (Fig. 6).
Bloomer et al. (2009) reported average $O_3$-temperature relationships of 2.2 – 2.4 $ppb_v$/°C for the Northeast, Southeast,
and Great Lakes regions of the U.S. across all $O_3$ percentiles. In contrast, the Southwest region, including Colorado,
had an average relationship of 1.4 $ppb_v$/°C (Bloomer et al., 2009). We find that $O_3$ was indeed correlated with
temperature at all NFRMA sites, with relationships that ranged from 0.07 to 1.95 $ppb_v$/°C with an average of 1.0 ± 0.4
$ppb_v$/°C (Fig. 8) for all sites and years. Quantitatively, this temperature dependence was low relative to other U.S.
sites, consistent with previous findings that biogenic VOCs contribute to, but did not dominate the VOC reactivity in
the NFRMA (McDuffie et al., 2016;Abeleira et al., 2017). However, the six NFRMA sites exhibited significant
variability in the 5th, 50th, and 95th percentiles among the sites both within a given year and across years (Fig. 9). The





$5^{th}$ and $95^{th}$ $O_3$ percentiles showed greater variability and larger uncertainties in the slopes than the $50^{th}$ percentile.
This indicated that baseline $O_3$ and high $O_3$ events in the region were less dependent on temperature. Baseline $O_3$ was
likely tied to the transport of $O_3$ and $O_3$ precursors from the west coast (Cooper et al., 2012), while the high $O_3$ events
were likely tied to a combination of meteorological parameters, including 500 hPa heights and stagnation events
(Reddy and Pfister, 2016), and local, temperature independent VOC emissions. In contrast, the $50^{th}$ percentile showed
a clear temperature dependence at all sites in most years (Fig. 8, Fig. 9), indicating that mean $O_3$ was typically
influenced by local temperature dependent, and likely biogenic, VOC emissions.
Unlike ambient $O_3$ and the weekend to weekday $\Delta O_3$, we noted no clear long-term trend in the $O_3$-temperature
relationship. The $O_3$-temperature relationships showed similar interannual patterns for the six sites at the $50^{th}$
percentile, except for years 2000-2003 (Fig. 9). Specifically, years 2001-2002, 2008, and 2011-2012 have suppressed
$O_3$-temperature slopes for the $50^{th}$ percentile. Reddy and Pfister (2016) reported high 500 hPa heights and $O_3$ for 2002-
2003, 2006, and 2012 while 2004 and 2009 had low 500 hPa heights and low $O_3$, so those exceptional years cannot
be explained solely by meteorology. However, those exceptional years (2002-2003, 2008, and 2011-2012) did
correspond to years in which Colorado was in moderate-severe drought with little soil moisture (NOAA, 2017).
Drought in the NFRMA is connected to changes in mountain-plains circulation and lower surface moisture, which
reduces the surface latent heat flux and causes increased surface temperature. These increased surface temperatures
lead to strong mountain-plains circulation, stagnant wind conditions, higher PBLs, and 500 hPa heights, all of which
are known to correlate with high $O_3$ episodes (Reddy and Pfister, 2016;Ek and Holtslag, 2004;Zhou and Geerts, 2013).
Drought is also connected to reduced isoprene emissions (Brilli et al., 2007;Fortunati et al., 2008;Guenther, 2006).
Consistent with this concept, Abeleira et al. (2017) noted that isoprene was 4 times higher at the Boulder Atmospheric
Observatory site in summer 2015 (a non-drought year) than in summer 2012 (a drought year). Such a decrease in
biogenic isoprene emissions should also suppress the $O_3$-temperature dependence in $NO_x$-saturated regimes, a trend
that was observed in the NFRMA (Fig. 9).
The suppressed $O_3$-temperature relationship during drought years in the NFRMA demonstrated the importance of
temperature dependent VOCs in driving $P(O_3)$ in the region, particularly at the mid-range $50^{th}$ percentile – but not at
the baseline $5^{th}$ percentile. A standard t-test showed that the $50^{th}$ and $95^{th}$ percentile slopes (i.e. temperature dependence
of average and high $O_3$ concentrations) are indeed different between the drought and non-drought years at the 95%
confidence limit. If $NO_x$ emissions continue to decrease, and the NFRMA continues its trend towards a $NO_x$-limited
regime (Fig. 7), the $O_3$-temperature dependence should also decrease and temperature-dependent VOCs will play a
smaller role in driving $O_3$ production. However, this would require substantial decreases in $NO_x$ for the heavily
trafficked Denver to become fully $NO_x$-limited, so temperature-dependent VOCs will likely remain important in at
least some regions of the NFRMA.
**4. Conclusions**
$O_3$ was decreasing across most of the country as $NO_x$ and VOC emissions continue to be reduced, with the exception
of background $O_3$ in the west (Cooper et al., 2012). In contrast, five out of six sites in the NFRMA showed no change
or increasing $O_3$ at the $50^{th}$ and $95^{th}$ percentiles between 2000 and 2015. While $NO_x$ levels have been reduced at the
CAMP and Welby sites in Denver, anthropogenic VOC emission estimates have increased as a result of increased
petroleum related activities (Fig. 4). A weekend-weekday analysis demonstrated that most sites in the NFRMA were
$NO_x$-saturated, but are transitioning to, and in one case may already have reached, the peak $P(O_3)$ cross-over point
between $NO_x$-saturated and $NO_x$-limited regimes. Some of the more rural NFRMA sites may already be in or near a
$NO_x$-limited system. This transition suggests that increasing anthropogenic VOC emissions will have less of an effect
on $P(O_3)$ in the region if $NO_x$ reductions continue, though VOCs still remain the limiting reagent for ozone production
in most of the NFRMA sites in 2015. Thus, the combined factors of increasing anthropogenic VOC emissions and
decreasing $NO_x$ in a $NO_x$-saturated system are likely culprits for the increasing $O_3$ trends within the NFRMA over the
past 15 years. Although the median $NO_2^*$ has decreased at the CAMP site from 37 $ppb_v$ in 2003 to 13 $ppb_v$ in 2015,
the site remains on the steep transitional part of the $P(O_3)$ curve between $NO_x$-saturated and peak $P(O_3)$ chemistry
(Fig. 6). Continued reductions in $NO_x$ emissions alone could lead to increased $O_3$ in the downtown Denver area until
the $P(O_3)$ chemistry passed the peak production region, although concurrent reductions in VOCs could mitigate the
increase in $P(O_3)$. As sources of VOCs and $NO_x$ change in the NFRMA with increased population, growth in the oil





and gas sector, and changing emissions regulations, continued analysis of $O_3$ and $NO_x$ will be essential for
understanding the shifting $P(O_3)$ regime. However, such analyses would benefit greatly from long-term $NO_x$
measurements at additional sites in the NFRMA.
$O_3$ in the NFRMA exhibits temperature dependence at all sites, but with varying intensities for different years. The 5th
and 95th $O_3$ percentiles demonstrated significant variability in temperature dependence for different sites in the same
year and across the study period, indicating that high $O_3$ events and background $O_3$ have other important controlling
factors such as transport of long-lived $O_3$ precursors from the west or meteorological parameters. Three time periods
exhibit a suppressed $O_3$-temperature dependence (2002-2003, 2008, and 2011-2012), coinciding with moderate to
extreme drought conditions in the NFRMA. These observations are consistent with the hypothesis that long-term
drought stress reduces biogenic VOC emissions and suppresses the $O_3$-temperature dependency. Climate change is
predicted to increase temperatures and thus increase $O_3$ by $1 - 10$ $ppb_v$ on a national scale (Jacob and Winner, 2009).
However, climate change models predict more extreme precipitation events in many areas, and estimates for Colorado
and the intermountain west suggest that drought may become more common in the region (Change, 2014). Our work
suggests that drought can temporarily suppress the $O_3$-temperature penalty in the NFRMA and potentially other $NO_x$-
saturated regions by reducing temperature dependent biogenic VOC emissions.

**Acknowledgements**
We thank the National Oceanic and Atmospheric Administration for funding this work (Award# NA14OAR4310148).



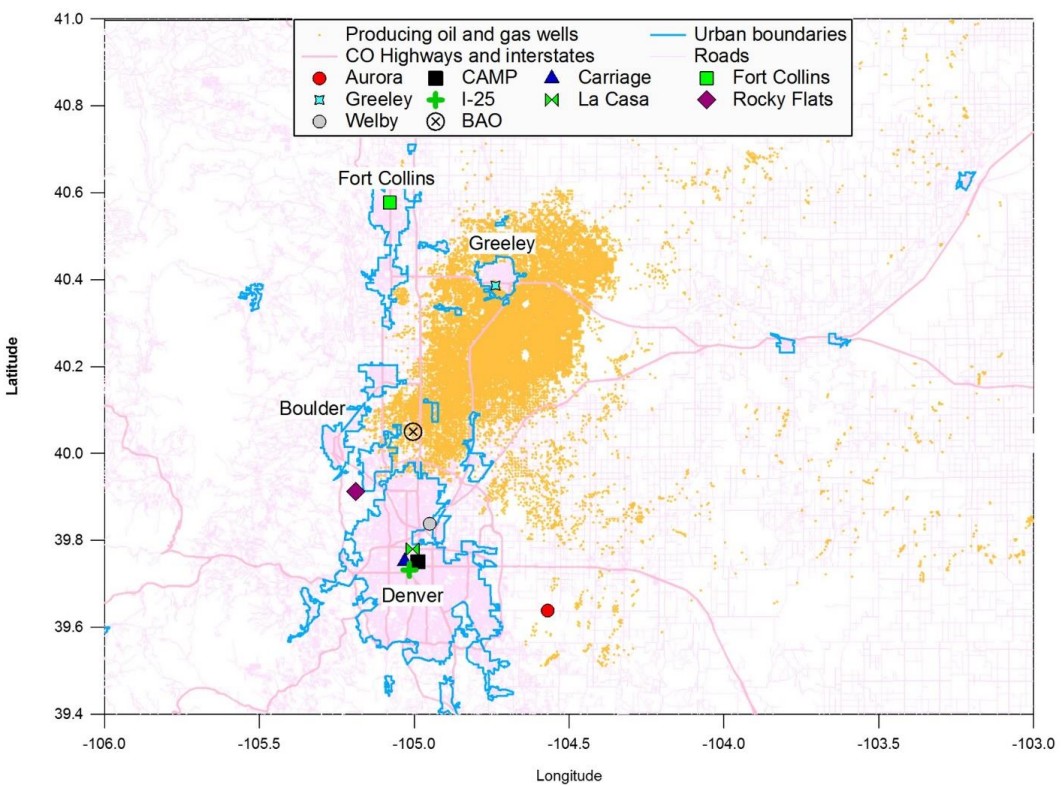

Figure 1. Site map for $O_3$ and $NO_2$ measurements in the NFRMA identified by shapes and colors. Producing oil and gas wells as of 2012 are identified on the map with gold dots. Urban areas are outlined with thick light-blue lines. Major interstates and state highways are identified by thick pink lines.





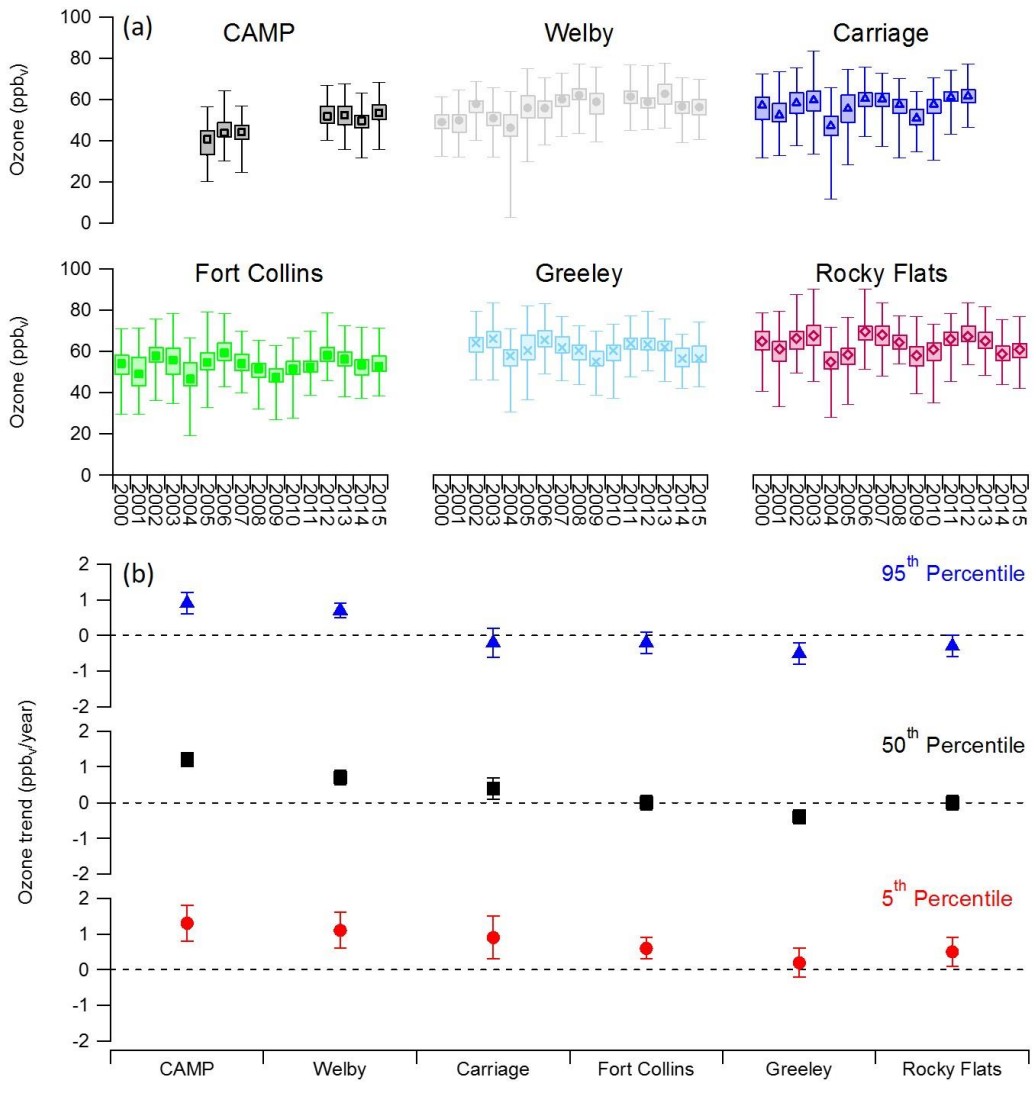

Figure 2. a) Trends in summer (June 1 – August 31) daytime (10:00 am – 4:00 pm) O₃ for six sites in the NFRMA between 2000 and 2015. Whiskers correspond to 5th and 9th percentiles, box thresholds correspond to 33rd and 67th percentiles, and the marker corresponds to the 50th percentile. Percentiles were determined from hourly daytime O₃ measurements at each site. The number of days used for each year's statistics depended on available data (n = 64 – 92). b) O₃ temporal trends were determined as the slope from annual trends (ppbᵥ O₃/year) from simple one-sided linear regression for the six NFRMA sites for the 95th (blue triangles), 50th (black squares), and 5th (red circles) percentiles. Error bars were calculated from the regression slope at one standard deviation.



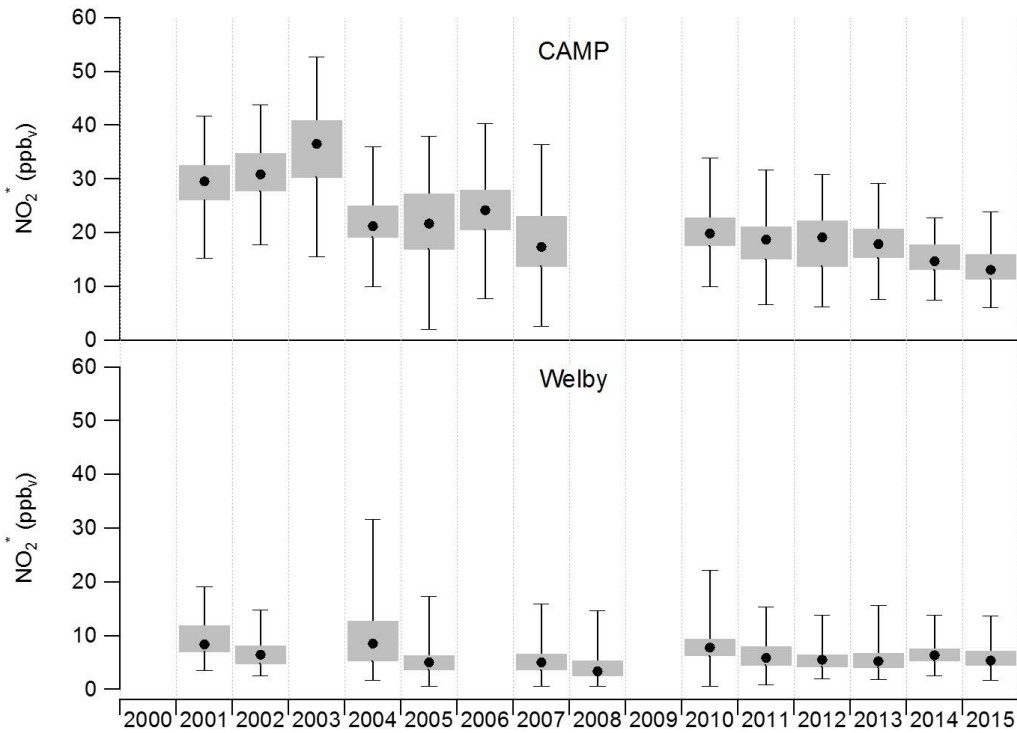

Figure 3. Box and whisker plots of NO$_2$* for the CAMP and Welby sites in Denver for all available data from 2000 – 2015. Whiskers correspond to 5[th] and 95[th] percentiles, box thresholds correspond to 33[rd] and 67[th] percentiles, and the black marker corresponds to the 50[th] percentile.





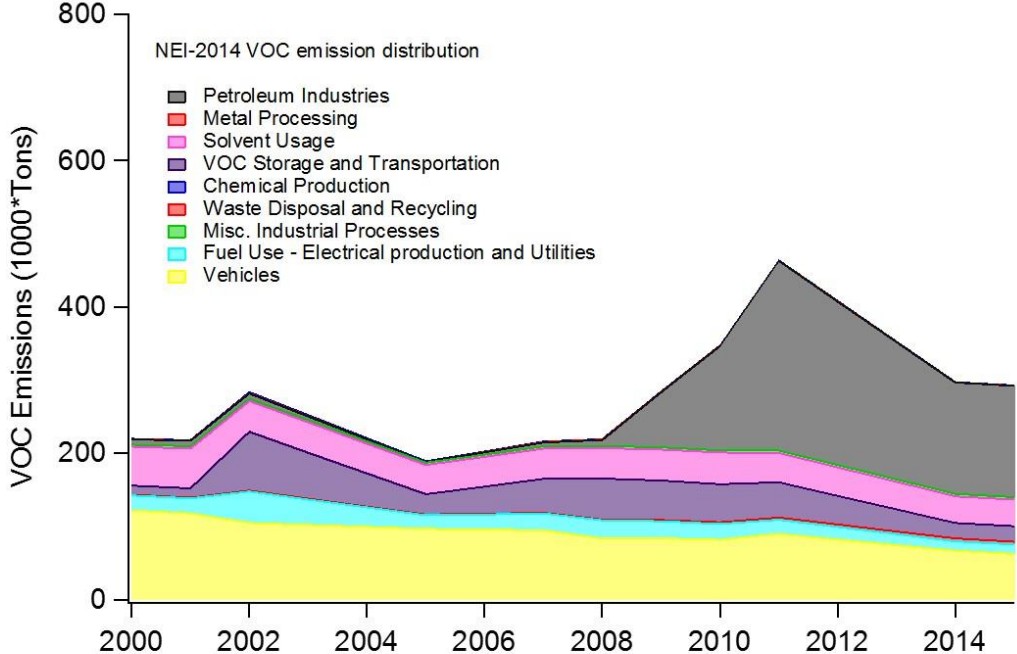

Figure 4: VOC emission estimates from EPA National Emissions Inventory 2014 (NEI-2014) for Colorado. Emission sources are separated by color, and are added to give the total VOC emission estimates for anthropogenic VOCs. Biogenic VOCs and VOCs from biomass burning (controlled fires and wildfires) are not included.





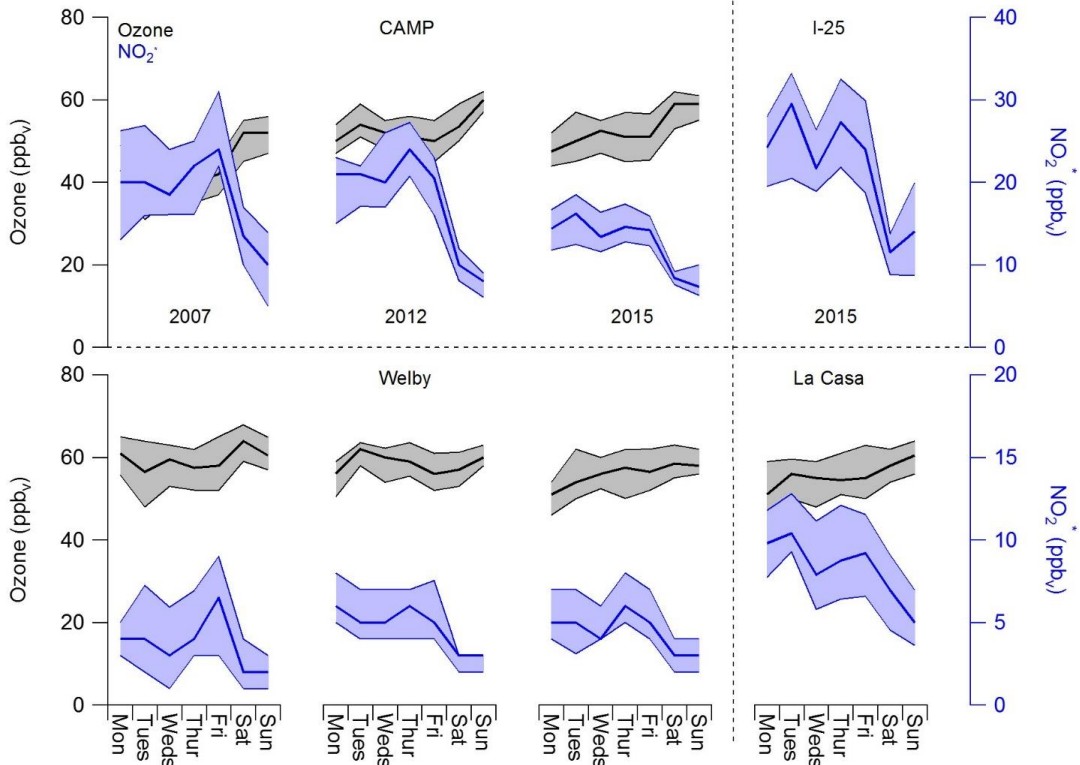

Figure 5. O₃ and NO₂* as a function of day of week for the CAMP, Welby, La Casa, and I-25 sites in Denver. All sites have plots for 2015, but only CAMP and Welby are plotted for 2007 and 2012 due to data availability. Solid lines are the 50th percentile for daytime hourly NO₂* (blue) and O₃ (black) measurements. The shaded regions are bounded by the 67th and 33rd percentiles. Note that the NO₂* y-axis scale is different on the upper and lower panels.





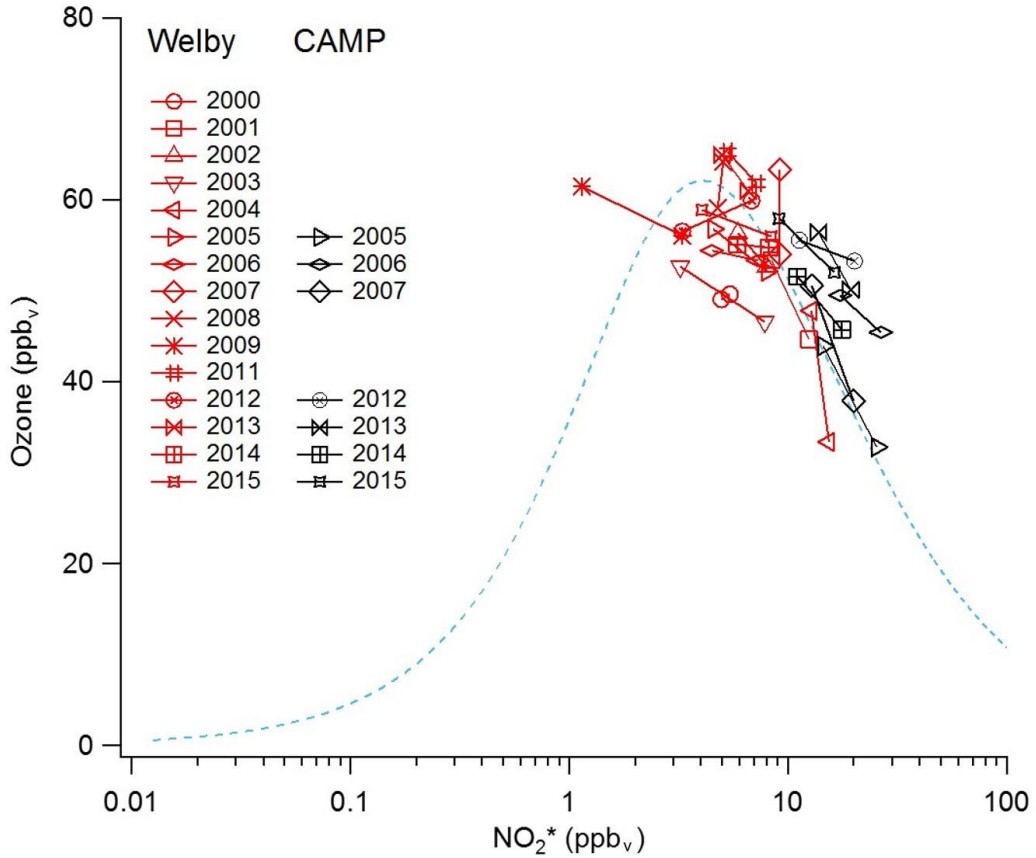

Figure 6. Weekday and weekend $O_3$ versus $NO_2$* for Welby (red) and CAMP (black) sites. Tethered symbols correspond to averages of Wednesday values for weekdays, and average Sunday values for weekends for each year depending on data availability. Standard errors of means for each year are <4 $ppb_v$ for $O_3$ and <2 $ppb_v$ for $NO_2$*. The dashed blue line is a visual aid to guide the readers eye to the non-linear $O_3$ curve, and was generated from the simple analytic model described by Farmer et al. (2011).





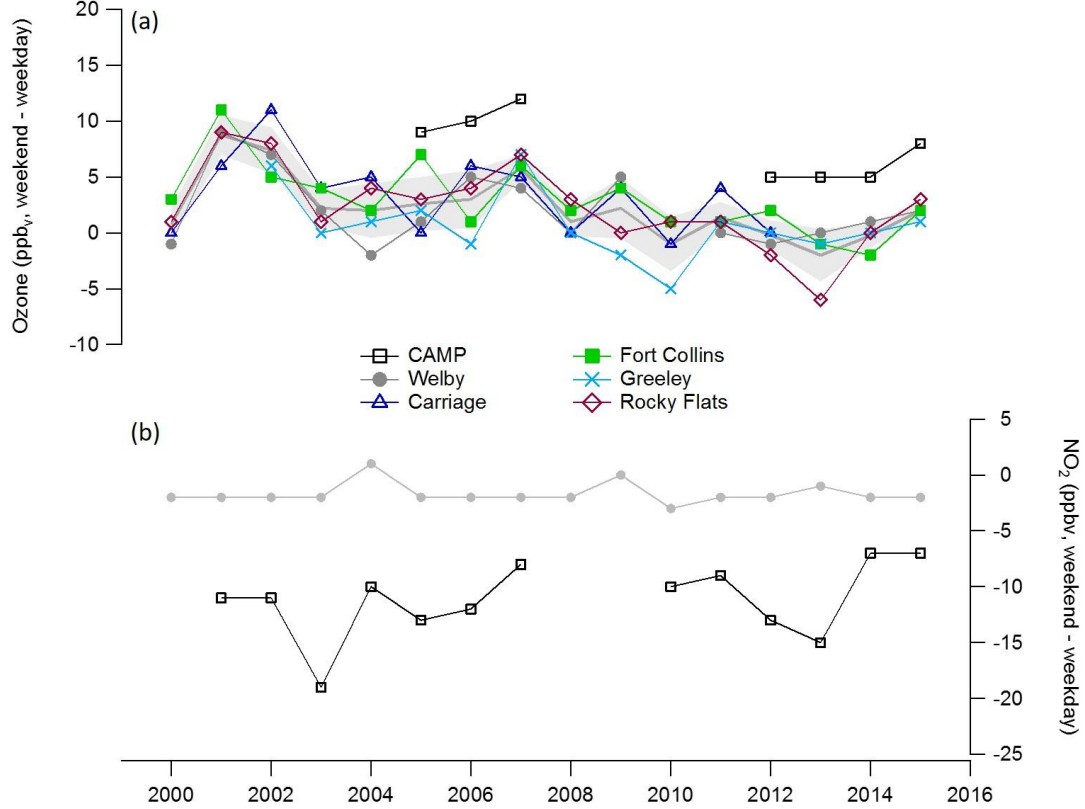

Figure 7: (a) The change in $O_3$ calculated as median weekend (Saturday to Sunday) minus summer weekday (Tuesday to Thursday) for the six NFRMA sites identified by color and marker for each year of available data. The solid grey line is the average median of the sites with the exception of CAMP. The inclusion of a site in the averaging for a given year was dependent on available data for that year. The light grey shading represents ± 1 standard deviation of the five site average. (b) The change in $NO_2$* calculated as median summer weekend (Saturday to Sunday) minus summer weekday (Tuesday to Thursday) for the CAMP and Welby sites.





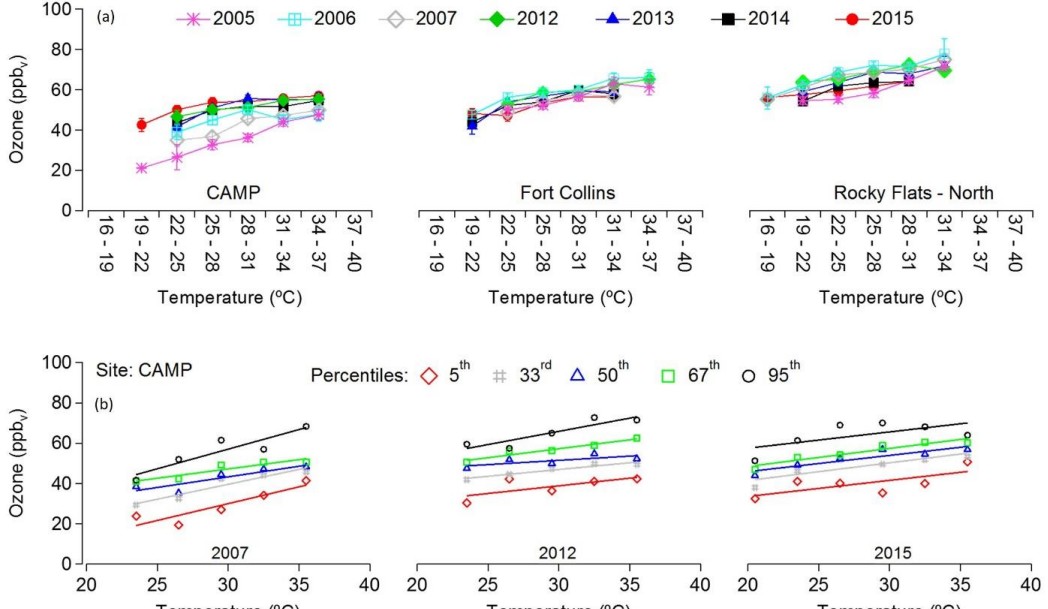

Figure 8. a) $O_3$ versus temperature for CAMP, Fort Collins, and Rocky Flats. $O_3$ is binned into 3°C temperature bins. Markers and colors represent yearly averages for each site. Error bars represent ± 1 standard error of the mean. Years were selected based on availability of overlapping data for multiple sites. b) One-sided linear regressions of 5°C temperature bins for 5th (red open diamond), 33rd (grey hash), 50th (blue open triangle), 67th (green open square), and 95th (black open circle) percentiles for the CAMP site for 2007 (left), 2012 (middle), and 2015 (right).





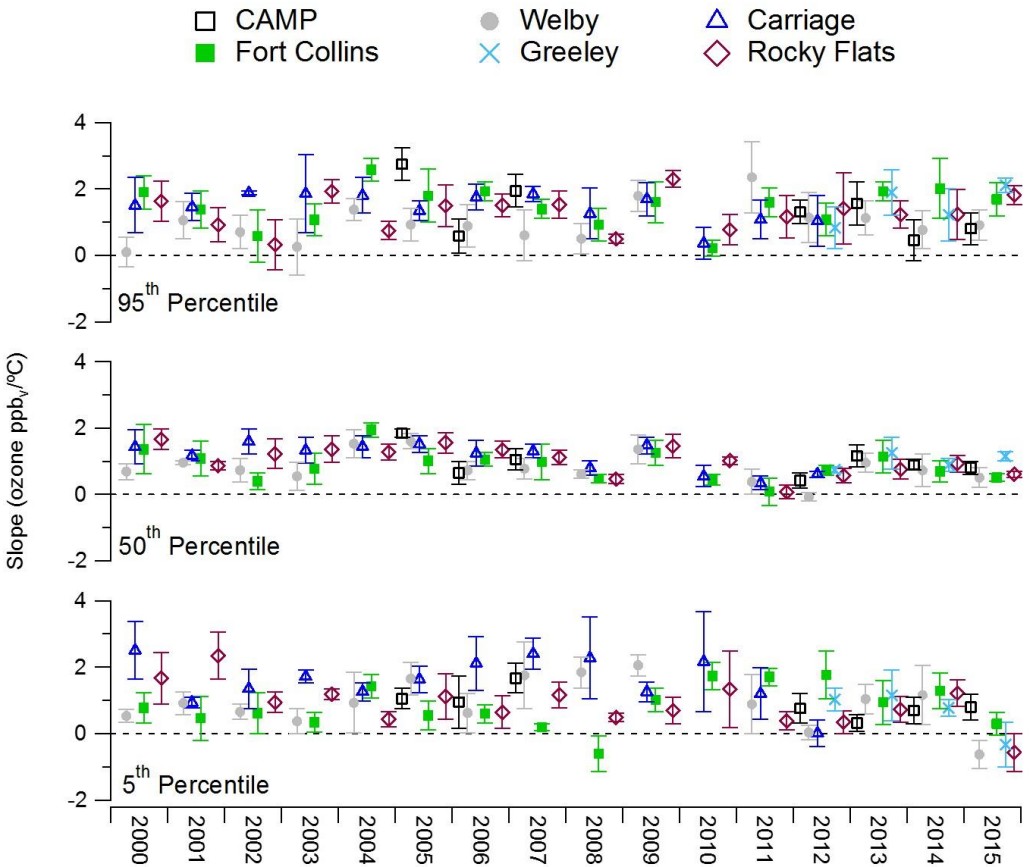

Figure 9. Slopes from one-sided linear regression of $O_3$ versus temperature (i.e. the temperature dependence of $O_3$) are binned into 5° Celsius bins for daytime (10:00 am – 4:00 pm) data at the 5th, 50th, and 95th percentiles for $O_3$. Data are shown for CAMP (black squares), Welby (grey solid circles), Carriage (blue open triangles), Fort Collins (green solid squares), Greeley (teal X's), and Rocky Flats (magenta open diamonds). Shaded years correspond to Colorado summers with moderate to severe drought conditions. Error bars are ±1 standard deviation of the slopes.





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

| Site | Latitude | Longitude | Elevation (m) | Measurements |
|------|----------|-----------|---------------|--------------|
| CAMP | 39.7512 | -104.988 | 1591 | $O_3$ & $NO_2^*$ |
| Welby | 39.8382 | -104.955 | 1554 | $O_3$ & $NO_2^*$ |
| Carriage | 39.7518 | -105.031 | 1619 | $O_3$ |
| Fort Collins | 40.5775 | -105.079 | 1523 | $O_3$ |
| Greeley | 40.3864 | -104.737 | 1476 | $O_3$ |
| Rocky Flats | 39.9128 | -105.189 | 1784 | $O_3$ |
| I-25 | 39.7321 | -105.015 | 1586 | $NO_2^*$ |
| La Casa | 39.7795 | -105.005 | 1601 | $O_3$ & $NO_2^*$ |

Table 1. Summary of Measurements sites used in this analysis. Note that $NO_2^*$ refers to the $NO_2$ detected by the EPA reference method, and thus includes a fraction of $NO_y$ species.