# Peer review of "Summer ozone in the Northern Front Range Metropolitan Area: Weekend-weekday effects, temperature dependences and the impact of drought"

_Atmospheric Chemistry and Physics, 2017_

## Author Comment (AC1) · 23 Feb 2017

Hello,

We have realized that Figure 9 is missing shading bars for drought years 2002-2003, 2008, 2011-2012 during which the ozone-temperature relationship is suppressed. We are uploading a revised Figure with the shading.

Thank you.
* * *
[Figure]

**Fig. 1.** Corrected Figure 9

---

## Referee Comment (RC1) · Anonymous Referee #1 · 31 Mar 2017

The manuscript investigates ozone formation in Colorado, a region that consistently exceeds the 8-hour ozone standard. The authors' find that the region is transitioning to a NOx-limited regime, as well as observe temperature dependencies of ozone attributed to drought. Overall, I found this manuscript to be very informative and straightforward, and timely for a region that is relatively less-studied than other areas of the country. The manuscript is well-written and figures clear. Most of my comments are minor and relate to clarity. With minor revisions, I recommend this manuscript for publication in ACP.

[Figure]

General Comments

(1) There is inconsistency in the statistics used. Figures 2 and 3 show 5th and 95th percentiles, while later figures show one siqma. Sometimes the standard deviation of the sample is shown (e.g., Figure 7) and other times the standard error of the mean (e.g., Figure 8). For clarity, I believe the authors should maintain consistency throughout the manuscript, and at a 95% confidence interval, needed to assess the statistical significance of results.

Specific Comments

(2) Lines 78-31. Do the authors mean *1980-1993* instead of "1980-2008"? Also, the ratio of VOC/NOx emissions has evolved with time in cities (Parrish et al., 2011; McDonald et al., 2013), which could also affect ozone trends.

Parrish, D. D., H. B. Singh, L. Molina, and S. Madronich (2011), Air quality progress in North American megacities: A review, Atmos Environ, 45, 7015-7025, doi:10.1016/j.atmosenv.2011.09.039.

McDonald, B. C., D. R. Gentner, A. H. Goldstein, and R. A. Harley (2013), Long-term trends in motor vehicle emissions in U.S. urban areas, Environ Sci Technol, 47, 10022-10031, doi:10.1021/es401034z.

(3) Line 164. This is an example where I found the inconsistency in statistics confusing. The error bars shown would suggest that all these results are statistically significant, rather than only at the 95th percentile.

(4) Lines 174 – 178. The authors' qualify the AVOC emissions trend shown in Figure 4 as an inventory estimate. I think this paragraph could be strengthened by referencing studies that have assessed emission trends for key sectors of this analysis, e.g., motor vehicles (e.g., McDonald et al., 2013), and oil and gas (e.g., Duncan et al., 2016), as well as studies that have reported uncertainties in emissions (e.g., Petron et al., 2014). What explains the hump in VOC emissions from petroleum industries around 2011? Is

this realistic, and comport with oil and natural gas production statistics from the Energy Information Administration? Such a rapid increase and decrease in VOC emissions would likely have some influence on observed ozone, as many of the points shown in Figure 6 are still on the NOx-saturated side of the curve. Also, McDuffie et al. (2016) suggested that maximum O3 was sensitive to NOx and reductions in VOCs in the Front Range.

Duncan, B. N., L. N. Lamsal, A. M. Thompson, Y. Yoshida, Z. F. Lu, D. G. Streets, M. M. Hurwitz, and K. E. Pickering (2016), A space-based, high-resolution view of notable changes in urban NOx pollution around the world (2005-2014), J Geophys Res-Atmos, 121, 976-996, doi:10.1002/2015jd024121.

Petron, G., A. Karion, C. Sweeney, B. R. Miller, S. A. Montzka, G. J. Frost, M. Trainer, P. Tans, A. Andrews, J. Kofler, D. Helmig, D. Guenther, E. Dlugokencky, P. Lang, T. Newberger, S. Wolter, B. Hall, P. Novelli, A. Brewer, S. Conley, M. Hardesty, R. Banta, A. White, D. Noone, D. Wolfe, and R. Schnell (2014), A new look at methane and nonmethane hydrocarbon emissions from oil and natural gas operations in the Colorado Denver-Julesburg Basin, J Geophys Res-Atmos, 119, 6836-6852, doi:10.1002/2013jd021272.

McDuffie, E. E., P. M. Edwards, J. B. Gilman, B. M. Lerner, W. P. Dube, M. Trainer, D. E. Wolfe, W. M. Angevine, J. De Gouw, E. J. Williams, A. G. Tevlin, J. G. Murphy, E. V. Fischer, S. McKeen, T. B. Ryerson, J. Peischl, J. Holloway, K. Aikin, A. O. Langford, C. J. Senff, R. J. Alvarez II, S. R. Hall, K. Ullmann, K. O. Lantz, and S. S. Brown (2016), Influence of oil and gas emissions on summertime ozone in the Colorado Northern Front Range, J Geophys Res-Atmos, 121, doi:10.1002/2016JD025265.

(5) Line 172. The NEI is reported for a single year. I believe the authors mean the EPA Trends Report, which is now reported by state.

(6) Line 187. The weekday/weekend effect is really due to a drop-off in heavy-duty truck traffic (Marr et al., 2002; McDonald et al., 2014). Passenger cars drive similar

amounts on weekends and weekdays.

Marr, L. C., and R. A. Harley (2002), Modeling the effect of weekday-weekend differences in motor vehicle emissions on photochemical air pollution in central California, Environ Sci Technol, 36, 4099-4106, doi:10.1021/Es020629x.

McDonald, B. C., Z. C. McBride, E. W. Martin, and R. A. Harley (2014), High-resolution mapping of motor vehicle carbon dioxide emissions, J Geophys Res-Atmos, 119, 5283-5298, doi:10.1002/2013jd021219.

(7) Line 209. I found the variability in concentrations across days, as shown in Figure 5, distracting for discerning weekday-weekend effects. I think this figure could be made clearer by showing a mean and confidence interval of weekdays (Mon-Friday), and of weekend days (Sa/Su) combined. Also, I think 95th confidence intervals should be shown, to make it easier for the reader to discern statistical significance.

(8) Lines 212-213. Are these 24 hour averages or daytime averages? If it is the former, could nighttime chemistry affect the weekday-weekend effect?

(9) Line 226-227. This sentence is confusing. Suggest revising.

(10) Lines 281-287. On Line 283, I believe the authors mean *2002-03* instead of "2001-02". To my eye in Figure 9, it is clear that 2008 and 2011-12 are suppressed, but I found it harder to see for 2002-2003. For 2002-2003, it only looks like the Fort Collins and Welby sites are suppressed, and not the other locations.

Minor Comments

(11) Line 211. Terminology switches from "weekday-weekend" to "weekend-weekday". Suggest choosing one word ordering and sticking with it.

---

## Referee Comment (RC2) · Anonymous Referee #2 · 8 Apr 2017

The authors investigate $O_3$ trends in the Northern Front Range Metropolitan Area of Colorado, a region which has exhibited ongoing issues with $O_3$ exceedances in spite of significant reductions in $NO_x$ emissions. In addition to examining overall trends over time, the authors use weekday/weekend comparisons of $NO_x$ and $O_3$ to help explain features of local chemistry, and also compare $O_3$ vs. temperature over time. Overall this paper is clear, well-organized, and represents a solid, if incremental addition to the existing air-pollution literature. I recommend publication, following improvements in a few areas.

1. First, and most importantly, I have concerns over the authors' use of binned temperatures as a preliminary step to linear regression. While I understand that this methodology has been utilized for similar purposes in the past, there are clear statistical flaws related to the practice that should be addressed before these results can be considered robust. Specific issues in the context of this paper include the following:

   - At relatively small sample sizes (n = 64-92 per summer), terms such as "95th percentile" become somewhat problematic. Dividing this already thin sample size into even smaller $3°C$ temperature bins must have, I assume, resulted in some bins with observations in the single digits. What methodology was used to determine percentiles from such small sample sizes?

   - By choosing uniformly spaced bin widths (years, in the case of this paper's temporal analysis, and uniform $3°C$ temperature widths in the case of the $O_3/T$ comparisons) information regarding sample sizes within each bin is lost completely. A bin containing more observations clearly should be weighted more heavily than a bin with fewer, but as written I see no indication that this kind of weighting was performed. This issue will be especially consequential for the temperature bins, since the relatively sparse temperature extremes will be incorrectly given weights equal to those of the middle bins, most likely exaggerating the resulting slopes. See Wasco and Sharma, 2014 for a description of how evenly spaced bins can produce exaggerated slopes as a result of this bias. Two methods that could correct this bias are equal number bins (with variable temperature widths based on the frequency distribution) and quantile regression (Koenker and Bassett, 1978). I think either of these would be superior to the current "equal distance bin" approach, with quantile regression also having the benefit of simultaneously addressing the small sample size issue.
   Wasko C, Sharma A. Quantile regression for investigating scaling of extreme

precipitation with temperature. Water Resour Res 2014;50:3608–14.

Koenker R, Bassett Jr G. Regression Quantiles. Econometrica 1978;46:33–50.

Further examples of this technique applied specifically to similar air-quality questions may be found elsewhere in the literature.

2. Figure 6: While I appreciate the attempt to use many symbols to distinguish years, I think the end result just doesn't work. The dense area around 10 ppb $NO_2$ in particular is nearly impossible to interpret easily. I suggest either abandoning the symbols entirely, and using shaded dots to represent different years, or else zooming in on the data to create more whitespace in this concentrated region.

3. The usage of "standard deviation" in several figure captions seems unclear. For example, on Figure 9 it seems to suggest that this is a standard deviation of many regression slopes. Is this the standard error of a single regression? Was bootstrapping performed, leading to many regression coefficients?

---

## Author Response (AR1)

**We thank the reviewers for their helpful comments. We have addressed every comment, and believe the result is an improved manuscript. Reviewer comments are below, with our author responses indented and in bold.**

**Response to Reviewer 1:**

The manuscript investigates ozone formation in Colorado, a region that consistently exceeds the 8-hour ozone standard. The authors' find that the region is transitioning to a NOx-limited regime, as well as observe temperature dependencies of ozone attributed to drought. Overall, I found this manuscript to be very informative and straightforward, and timely for a region that is relatively less-studied than other areas of the country. The manuscript is well-written and figures clear. Most of my comments are minor and relate to clarity. With minor revisions, I recommend this manuscript for publication in ACP.

General Comments (1) There is inconsistency in the statistics used. Figures 2 and 3 show 5th and 95th percentiles, while later figures show one siqma. Sometimes the standard deviation of the sample is shown (e.g., Figure 7) and other times the standard error of the mean (e.g., Figure 8). For clarity, I believe the authors should maintain consistency throughout the manuscript, and at a 95% confidence interval, needed to assess the statistical significance of results.

**Thank you to the reviewer for pointing out the inconsistencies in the error reporting. We agree that this should be improved, and the figures and references to figures or data have been updated in the revised manuscript as follows:**

**Figure 2b. The error bars are now the 95% confidence intervals around the reported ozone/year slopes.**

**Figure 3b. We included an additional figure similar to Figure 2b to show the $NO_2$/year slopes for the 5th, 50th, and 95th percentiles with the error bars representing the 95% confidence intervals around the slopes.**

**Figure 5 was updated with suggestions from comment 7 to show the weekday and weekend averages with the 95% confidence intervals.**

**Figure 7a was updated and shows the average weekday minus weekend ozone for each year for the six sites. The solid grey line represents the aggregated average of the six sites with the shading representing the 95% confidence interval.**

**Figure 7b was updated and shows the average weekday minus weekend $NO_2$ for each year for the CAMP and Welby sites. The error bars now represent the 95% confidence interval of the averages.**

**Figures 8 and 9 were updated to include averages and 95% confidence intervals, and also to change the temperature binning approach as suggested by the second reviewer.**

**Figure 8a. was updated with the new equal bin size approach suggested by reviewer 2, and the averages of those temperature bins for each year are displayed. The 95% confidence**

**intervals for the O₃ bin averages were not included in the figure for clarity purposes, but are typically <8 ppbᵥ.**

**Figure 9 was updated with the new equal bin size approach suggested by reviewer 2, and the 95% confidence intervals around the yearly O₃/temperature slopes are included.**

Specific Comments (2) Lines 78-31. Do the authors mean *1980-1993* instead of "1980-2008"? Also, the ratio of VOC/NOx emissions has evolved with time in cities (Parrish et al., 2011; McDonald et al., 2013), which could also affect ozone trends. Parrish, D. D., H. B. Singh, L. Molina, and S. Madronich (2011), Air quality progress in North American megacities: A review, Atmos Environ, 45, 7015-7025, doi:10.1016/j.atmosenv.2011.09.039. McDonald, B. C., D. R. Gentner, A. H. Goldstein, and R. A. Harley (2013), Long-term trends in motor vehicle emissions in U.S. urban areas, Environ Sci Technol, 47, 10022-10031, doi:10.1021/es401034z.

> **We do mean "1980-2008". Lefohn et. al (2010) compare trends at monitoring sites across the US for two overlapping time periods 1980-2008 and 1994-2008. They found that many sites had a decreasing O₃ trend for the longer 1980-2008 period, but most of the decreasing trends were not present during the 1994-2008 period indicating that O₃ decreases had slowed or stopped in the 1994-2008 period. We have revised the statement to try and clarify that point.**

> > **Lefohn et al. (2010) found that when comparing O₃ at the same sites for a longer period of 1980-2008 and shorter period of 1994-2008, the predominant pattern was a change from a negative trend (decreasing O₃) during the longer period to no trend (stagnant O₃) in the shorter period, indicating that O₃ reductions had leveled off by the late 2000s.**

> **We thank the reviewer for their comment and suggestion and have including the following reference as suggested;**

> > **McDonald et al. (2013) report decreased VOC, CO, and NOₓ automobile emissions in major US urban centers, and more importantly decreasing VOC/NOₓ trends from 1990 to 2007 with a turnaround and small increase after 2007, which would affect local O₃ chemistry within the city and at downwind receptor sites**

(3) Line 164. This is an example where I found the inconsistency in statistics confusing. The error bars shown would suggest that all these results are statistically significant, rather than only at the 95th percentile.

> **In this section Figures 2 and 3 were referenced, both of which have been updated per comment 1. The statistical significance of the long-term O₃ and NO₂ trends were determine from both an F-test and from the 95% confidence intervals around the slope.**

(4) Lines 174 – 178. The authors' qualify the AVOC emissions trend shown in Figure 4 as an inventory estimate. I think this paragraph could be strengthened by referencing studies that have assessed emission trends for key sectors of this analysis, e.g., motor vehicles (e.g., McDonald et al., 2013), and oil and gas (e.g., Duncan et al., 2016), as well as studies that have reported uncertainties in emissions (e.g.,

Petron et al., 2014). What explains the hump in VOC emissions from petroleum industries around 2011? Is this realistic, and comport with oil and natural gas production statistics from the Energy Information Administration? Such a rapid increase and decrease in VOC emissions would likely have some influence on observed ozone, as many of the points shown in Figure 6 are still on the NOx-saturated side of the curve. Also, McDuffie et al. (2016) suggested that maximum O3 was sensitive to NOx and reductions in VOCs in the Front Range.

Duncan, B. N., L. N. Lamsal, A. M. Thompson, Y. Yoshida, Z. F. Lu, D. G. Streets, M. M. Hurwitz, and K. E. Pickering (2016), A space-based, high-resolution view of notable changes in urban NOx pollution around the world (2005-2014), J Geophys Res-Atmos, 121, 976-996, doi:10.1002/2015jd024121.

Petron, G., A. Karion, C. Sweeney, B. R. Miller, S. A. Montzka, G. J. Frost, M. Trainer, P. Tans, A. Andrews, J. Kofler, D. Helmig, D. Guenther, E. Dlugokencky, P. Lang, T. Newberger, S. Wolter, B. Hall, P. Novelli, A. Brewer, S. Conley, M. Hardesty, R. Banta, A. White, D. Noone, D. Wolfe, and R. Schnell (2014), A new look at methane and nonmethane hydrocarbon emissions from oil and natural gas operations in the Colorado Denver-Julesburg Basin, J Geophys Res-Atmos, 119, 6836-6852, doi:10.1002/2013jd021272.

McDuffie, E. E., P. M. Edwards, J. B. Gilman, B. M. Lerner, W. P. Dube, M. Trainer, D. E. Wolfe, W. M. Angevine, J. De Gouw, E. J. Williams, A. G. Tevlin, J. G. Murphy, E. V. Fischer, S. McKeen, T. B. Ryerson, J. Peischl, J. Holloway, K. Aikin, A. O. Langford, C. J. Senff, R. J. Alvarez II, S. R. Hall, K. Ullmann, K. O. Lantz, and S. S. Brown (2016), Influence of oil and gas emissions on summertime ozone in the Colorado Northern Front Range, J Geophys Res-Atmos, 121, doi:10.1002/2016JD025265.

**The reviewer's suggestion motivated us to include an updated Figure 4 to include the number of active oil and natural gas wells in Colorado from 2000 to 2015 and the yearly average natural gas withdrawal estimates from the Energy Information Administration, which show increases in both number of wells and the natural gas withdrawal in Colorado (see updated figure 4 below). We have included the following text for some more information regarding ONG in Colorado, changing VOC emissions around the country, and impacts on ozone in the Front Range.**

**The US Energy Information Administration (EIA) report a 2-fold increase in active ONG wells from ~25000 to ~40000 from 2010 to 2012 (Fig. 4c) (US-EIA, 2017). A number of VOC studies in the NFRMA since 2011 report enhanced $C_2$-$C_5$ alkanes relative to other urban/semi-urban regions (Abeleira et al., 2017;McDuffie et al., 2016;Pétron et al., 2012;Pétron et al., 2014;Swarthout et al., 2013). Pétron et al. (2014) reported that the state inventory for total VOCs emitted by ONG activities was at least 2x lower than May 2012, which indicates that the contribution of ONG related VOCs in figure 4 would increase substantially. McDonald et al. (2013) report decreases in both $NO_x$ and VOC emissions from automobiles, and a steady reduction in the VOC/$NO_x$ emission ratio in major cities from 1990 to 2008, with a possible trend reversal following 2008. McDuffie et al. (2016) reported that maximum $O_3$ at a site in the NFRMA was sensitive to $NO_x$ and VOC reductions.**

[Figure]

(5) Line 172. The NEI is reported for a single year. I believe the authors mean the EPA Trends Report, which is now reported by state.

**The reviewer is correct, and this mistake was revised in the manuscript. Thank you.**

(6) Line 187. The weekday/weekend effect is really due to a drop-off in heavy-duty truck traffic (Marr et al., 2002; McDonald et al., 2014). Passenger cars drive similar amounts on weekends and weekdays.

Marr, L. C., and R. A. Harley (2002), , Environ Sci Technol, 36, 4099-4106, doi:10.1021/Es020629x. McDonald, B. C., Z. C. McBride, E. W. Martin, and R. A. Harley (2014), High-resolution mapping of motor vehicle carbon dioxide emissions, J Geophys Res-Atmos, 119, 5283- 5298, doi:10.1002/2013jd021219.

**We agree with the reviewer, and have included the following revision;**

**Traffic patterns in urban regions are different between weekends and weekdays with a decrease in heavy-duty truck traffic on weekends (Marr and Harley, 2002). VOCs are expected to be stable across the week (Marr and Harley, 2002) as major VOC sources do not vary by day-of-week.**

(7) Line 209. I found the variability in concentrations across days, as shown in Figure 5, distracting for discerning weekday-weekend effects. I think this figure could be made clearer by showing a mean and confidence interval of weekdays (Mon-Friday), and of weekend days (Sa/Su) combined. Also, I think 95th confidence intervals should be shown, to make it easier for the reader to discern statistical significance.

**The suggestion from the reviewer was used to clarify the data presented on Figure 5. Figure 5 was remade with average +/- 95% confidence interval for the same sites and years as the original figure. See updated figure below.**

(8) Lines 212-213. Are these 24 hour averages or daytime averages? If it is the former, could nighttime chemistry affect the weekday-weekend effect?

**All data presented in the manuscript is constrained to daytime (10:00am – 4:00pm local) values.**

(9) Line 226-227. This sentence is confusing. Suggest revising.

**This section was updated with new insight provided by updating the figures, and includes the following revisions.**

**Measured $NO_2^*$ decreased at both CAMP and Welby between 2001 and 2015 (Fig. 3b), but with larger decreases at the CAMP site. The $\Delta NO_2^*$ at Welby remained stable with an average value of -1.7 ± 0.9 $ppb_v$, while $\Delta NO_2^*$ at the CAMP site exhibited a statistically significant decrease of 0.6 ± 0.4 $ppb_v$/yr. The decreasing $\Delta NO_2^*$ at the CAMP site appears to be converging with $\Delta NO_2^*$ at the Welby site. It is unlikely that traffic patterns are assimilating between the two sites, and a more plausible explanation is that emission control technologies on heavy duty commercial fleet vehicles are reducing the impact on emissions of those specific vehicles, and thus reducing the measurable $\Delta NO_2^*$ (Bishop et al., 2015).**

(10) Lines 281-287. On Line 283, I believe the authors mean *2002-03* instead of "2001-02". To my eye in Figure 9, it is clear that 2008 and 2011-12 are suppressed, but I found it harder to see for 2002-2003. For 2002-2003, it only looks like the Fort Collins and Welby sites are suppressed, and not the other locations.

**We have updated the manuscript to reflect this observation.**

Minor Comments (11) Line 211. Terminology switches from "weekday-weekend" to "weekend-weekday". Suggest choosing one word ordering and sticking with it.

**The terminology throughout the manuscript has been updated to "weekend-weekday".**

**Response to Reviewer 2:**

The authors investigate O3 trends in the Northern Front Range Metropolitan Area of Colorado, a region which has exhibited ongoing issues with O3 exceedances in spite of significant reductions in NOx emissions. In addition to examining overall trends over time, the authors use weekday/weekend comparisons of NOx and O3 to help explain features of local chemistry, and also compare O3 vs. temperature over time. Overall this paper is clear, well-organized, and represents a solid, if incremental addition to the existing air-pollution literature. I recommend publication, following improvements in a few areas.

First, and most importantly, I have concerns over the authors' use of binned temperatures as a preliminary step to linear regression. While I understand that this methodology has been utilized for similar purposes in the past, there are clear statistical flaws related to the practice that should be addressed before these results can be considered robust. Specific issues in the context of this paper include the following: • At relatively small sample sizes (n = 64-92 per summer), terms such as "95th percentile" become somewhat problematic. Dividing this already thin sample size into even smaller 3∘C temperature bins must have, I assume, resulted in some bins with observations in the single digits. What methodology was used to determine percentiles from such small sample sizes? • By choosing uniformly spaced bin widths (years, in the case of this paper's temporal analysis, and uniform 3∘C temperature widths in the case of the O3/T comparisons) information regarding sample sizes within each bin is lost completely. A bin containing more observations clearly should be weighted more heavily than a bin with fewer, but as written I see no indication that this kind of weighting was performed. This issue will be especially consequential for the temperature bins, since the relatively sparse temperature extremes will be incorrectly given weights equal to those of the middle bins, most likely exaggerating the resulting slopes. See Wasco and Sharma, 2014 for a description of how evenly spaced bins can produce exaggerated slopes as a result of this bias. Two methods that could correct this bias are equal number bins (with variable temperature widths based on the frequency distribution) and quantile regression (Koenker and Bassett, 1978). I think either of these would be superior to the current "equal distance bin" approach, with quantile regression also having the benefit of simultaneously addressing the small sample size issue. Wasko C, Sharma A. Quantile regression for investigating scaling of extreme precipitation with temperature. Water Resour Res 2014;50:3608–14. Koenker R, Bassett Jr G. Regression Quantiles. Econometrica 1978;46:33– 50. Further examples of this technique applied specifically to similar air-quality questions may be found elsewhere in the literature.

> **Thank you to the reviewer for a detailed explanation of the issues with uniformly spaced temperature bins, and the suggestion of weighting the yearly trends. We will address both topics below:**
>
> 1) **Temporal trends and weighting of years: The EPA ozone, NO$_2$, and temperature data are available at an hourly time resolution. For the temporal trends of ozone and NO$_2$ we calculated daily averages for 10:00 am – 4:00 pm for summer data (Jun-Aug). To determine the percentiles for each summer at a site we aggregated the daily averages and applied the Tukey method to find the 5th, 33rd, 50th, 67th, and 95th percentiles (figure 2a, figure 3a). As the reviewer noted relatively small sample sizes can be problematic when calculating high or low percentiles (95th and 5th). We believe that the reviewer is referring to the tendency for the percentile calculations at the 5th or 95th to be skewed by low and high outliers, which**

becomes more problematic as the sample size decreases. As the sample size becomes sufficiently small the 5th and 95th percentiles will tend to equal the minimum and maximum values of the data, which can be outliers. We went back through the yearly trends to investigate the influence of outliers on the percentiles and found that only 1 year at 2 sites (Welby and Carriage 2004) exhibited 1 day of unrealistically low ozone (<5 ppbv), which is lower than typical background ozone, and were removed as outliers to not skew the 5th percentile values. Below is a table summarizing the number of daily average points for each year used in the percentile calculations.

| | Number of points in long term ozone trend daily averages | | | | | | NO$_2$ trends | |
|------|-------|-------------|---------|--------------|----------|------|------|-------|
| Year | Welby | Rocky Flats | Greeley | Fort Collins | Carriage | CAMP | CAMP | Welby |
| 2000 | 90 | 88 | | 89 | 91 | | | |
| 2001 | 89 | 90 | | 91 | 90 | | 89 | 89 |
| 2002 | 88 | 85 | 87 | 91 | 91 | | 85 | 78 |
| 2003 | 86 | 91 | 91 | 91 | 91 | | 74 | |
| 2004 | 87 | 91 | 91 | 91 | 85 | | 80 | 81 |
| 2005 | 91 | 91 | 91 | 91 | 89 | 63 | 91 | 91 |
| 2006 | 90 | 91 | 91 | 91 | 88 | 91 | 82 | |
| 2007 | 91 | 89 | 91 | 91 | 86 | 90 | 89 | 91 |
| 2008 | 90 | 91 | 87 | 91 | 91 | | | 90 |
| 2009 | 84 | 91 | 91 | 91 | 91 | | | |
| 2010 | | 89 | 91 | 77 | 90 | | 91 | 78 |
| 2011 | 91 | 91 | 91 | 91 | 91 | | 71 | 86 |
| 2012 | 87 | 91 | 90 | 90 | 80 | 91 | 90 | 71 |
| 2013 | 86 | 90 | 91 | 75 | | 91 | 91 | 86 |
| 2014 | 91 | 91 | 90 | 78 | | 91 | 91 | 91 |
| 2015 | 90 | 91 | 91 | 91 | | 90 | 90 | 84 |

The reviewer suggests weighting the yearly trends by the number of data points to correct for differences in the number of points in different years. However, we note that >90% of the years for all sites with available data have 80-92 daily averages, and we thus expect a negligible effect on the analysis from weighting based on the number of data points.

2) Uniformly spaced temperature bins versus temperature bins with the same number of data points: The reviewer suggests redoing the ozone-temperature analysis using temperature bin widths dictated by a constant number of data points in a bin instead of using uniform temperature bins. As the reviewer noted we were dividing an already small sample size of 80-90 daily averages into temperature bins, some of which contained <10 data points for the high and low temperature bins. Applying the percentile calculations to such small sample sizes was not statistically robust, and tended to only yield the minimum and maximum values for those temperature bins. To increase the number of data points for a more robust statistical analysis we used the hourly ozone and temperature data. For a full 92-day summer data set we are now working with 552 data points (10:00am – 4:00pm, 6 hours per day). The 552 data points were split into 5 temperature bins with 110 data points each, with the two extra points disregarded. Due to missing data, the smallest number of data points for a single temperature bin was 51 (CAMP 2005), but >90% of bins contained 100-110 data points. Due to the scarcity of bins with <100 data points we did not weight the ozone-temperature relationships by the number of points in each bin. We have updated figures 8 and 9 with this improved analysis. Below are summary tables of the number of ozone points in each temperature bin for each site and year. We note that this has no substantive effect on the interpretation of the data, nor conclusions drawn, but does make for a more robust analysis.

**Number of Points in Welby temperature bins    Number of Points in Rocky Flats temperature bins    Number of Points in Greeley temperature bins**

| Year | Bin 1 | Bin 2 | Bin 3 | Bin 4 | Bin 5 | Bin 1 | Bin 2 | Bin 3 | Bin 4 | Bin 5 | Bin 1 | Bin 2 | Bin 3 | Bin 4 | Bin 5 |
|------|-------|-------|-------|-------|-------|-------|-------|-------|-------|-------|-------|-------|-------|-------|-------|
| 2000 | 104 | 110 | 110 | 110 | 110 | 103 | 105 | 107 | 107 | 110 | | | | | |
| 2001 | 106 | 108 | 109 | 105 | 110 | 107 | 107 | 108 | 110 | 110 | | | | | |
| 2002 | 105 | 106 | 107 | 109 | 102 | 102 | 98 | 99 | 96 | 101 | | | | | |
| 2003 | 97 | 96 | 104 | 110 | 106 | 109 | 104 | 110 | 109 | 109 | | | | | |
| 2004 | 96 | 108 | 105 | 105 | 104 | 107 | 109 | 108 | 108 | 105 | | | | | |
| 2005 | 108 | 107 | 110 | 110 | 109 | 110 | 110 | 108 | 110 | 110 | | | | | |
| 2006 | 109 | 105 | 106 | 109 | 100 | 109 | 109 | 108 | 107 | 110 | | | | | |
| 2007 | 110 | 110 | 110 | 108 | 108 | 110 | 107 | 108 | 109 | 98 | | | | | |
| 2008 | 104 | 103 | 106 | 110 | 109 | 107 | 110 | 105 | 110 | 110 | | | | | |
| 2009 | 102 | 93 | 99 | 92 | 103 | 109 | 110 | 109 | 109 | 109 | | | | | |
| 2010 | | | | | | 110 | 108 | 102 | 103 | 96 | | | | | |
| 2011 | 109 | 107 | 105 | 108 | 110 | 106 | 110 | 110 | 110 | 110 | | | | | |
| 2012 | 106 | 106 | 110 | 110 | 62 | 110 | 110 | 110 | 108 | 108 | 110 | 109 | 109 | 109 | 107 |
| 2013 | 110 | 109 | 106 | 108 | 72 | 106 | 110 | 110 | 110 | 105 | 110 | 110 | 103 | 108 | 109 |
| 2014 | 110 | 110 | 109 | 110 | 109 | 110 | 110 | 110 | 110 | 110 | 108 | 109 | 108 | 108 | 104 |
| 2015 | 103 | 108 | 110 | 107 | 109 | 107 | 110 | 110 | 110 | 108 | 108 | 105 | 108 | 108 | 108 |

**Number of Points in Fort Collins temperature bins    Number of Points in Carriage temp bins    Number of Points in Camp temp bins**

| Year | Bin 1 | Bin 2 | Bin 3 | Bin 4 | Bin 5 | Bin 1 | Bin 2 | Bin 3 | Bin 4 | Bin 5 | Bin 1 | Bin 2 | Bin 3 | Bin 4 | Bin 5 |
|------|-------|-------|-------|-------|-------|-------|-------|-------|-------|-------|-------|-------|-------|-------|-------|
| 2000 | 104 | 109 | 108 | 107 | 107 | 90 | 94 | 95 | 90 | 91 | | | | | |
| 2001 | 77 | 90 | 91 | 93 | 96 | 109 | 103 | 109 | 109 | 109 | | | | | |
| 2002 | 81 | 88 | 98 | 93 | 72 | 105 | 108 | 110 | 109 | 110 | | | | | |
| 2003 | 107 | 106 | 107 | 109 | 104 | 106 | 105 | 110 | 109 | 110 | | | | | |
| 2004 | 110 | 110 | 108 | 110 | 105 | 109 | 109 | 108 | 108 | 108 | | | | | |
| 2005 | 70 | 89 | 102 | 108 | 108 | 109 | 105 | 104 | 101 | 94 | 51 | 74 | 70 | 74 | 103 |
| 2006 | 107 | 107 | 110 | 110 | 110 | 92 | 109 | 109 | 104 | 105 | 110 | 109 | 108 | 107 | 107 |
| 2007 | 109 | 107 | 108 | 108 | 110 | 106 | 98 | 105 | 109 | 110 | 108 | 104 | 108 | 109 | 110 |
| 2008 | 109 | 109 | 108 | 107 | 110 | 90 | 103 | 104 | 100 | 107 | | | | | |
| 2009 | 105 | 110 | 110 | 109 | 110 | 107 | 110 | 109 | 110 | 110 | | | | | |
| 2010 | 104 | 110 | 110 | 110 | 110 | 109 | 110 | 109 | 110 | 110 | | | | | |
| 2011 | 110 | 110 | 108 | 108 | 110 | 108 | 106 | 109 | 110 | 104 | | | | | |
| 2012 | 110 | 108 | 105 | 108 | 100 | 108 | 108 | 110 | 110 | 108 | 108 | 107 | 109 | 110 | 109 |
| 2013 | 110 | 108 | 108 | 109 | 109 | | | | | | 108 | 107 | 110 | 110 | 109 |
| 2014 | 109 | 110 | 110 | 110 | 110 | | | | | | 109 | 110 | 110 | 110 | 110 |
| 2015 | 95 | 108 | 110 | 109 | 105 | | | | | | 110 | 110 | 109 | 108 | 105 |

2. Figure 6: While I appreciate the attempt to use many symbols to distinguish years, I think the end result just doesn't work. The dense area around 10 ppb NO2 in particular is nearly impossible to interpret easily. I suggest either abandoning the symbols entirely, and using shaded dots to represent different years, or else zooming in on the data to create more whitespace in this concentrated region.

**We have revised this figure to minimize the visual interference and clustering of the symbols. The revised figure is below:**

[Figure]

3. The usage of "standard deviation" in several figure captions seems unclear. For example, on Figure 9 it seems to suggest that this is a standard deviation of many regression slopes. Is this the standard error of a single regression? Was bootstrapping performed, leading to many regression coefficients?

**We have revised and updated most of the figures per a suggestion from reviewer 1 to be more consistent with the error analysis. The updates are as follows;**

**Figure 2b. The error bars are now the 95% confidence intervals around the reported ozone/year slopes.**

**Figure 3b.** We included an additional figure similar to Figure 2b to show the $NO_2$/year slopes for the 5th, 50th, and 95th percentiles with the error bars representing the 95% confidence intervals around the slopes.

**Figure 5** was updated with suggestions from reviewer 1 comment 7 to show the weekday and weekend averages with the 95% confidence intervals.

**Figure 7a** was updated and shows the average weekday minus weekend ozone for each year for the six sites. The solid grey line represents the aggregated average of the six sites with the shading representing the 95% confidence interval.

**Figure 7b** was updated and shows the average weekday minus weekend $NO_2$ for each year for the CAMP and Welby sites. The error bars represent the 95% confidence interval of the averages.

**Figure 8a** was updated with the new equal bin size approach, and the averages of those temperature bins for each year are displayed. The 95% confidence intervals for the $O_3$ bin averages were not included in the figure for clarity purposes, but are typically <8 $ppb_v$.

**Figure 9** was updated with the new equal bin size approach suggested, and the 95% confidence intervals around the yearly $O_3$/temperature slopes are included.

Abeleira, A., Pollack, I., Sive, B. C., Zhou, Y., Fischer, E. V., and Farmer, D.: Source Characterization of Volatile Organic Compounds in the Colorado Northern Front Range Metropolitan Area during Spring and Summer 2015, Journal of Geophysical Research, In Press, 2017.

Bishop, G. A., Hottor-Raguindin, R., Stedman, D. H., McClintock, P., Theobald, E., Johnson, J. D., Lee, D.-W., Zietsman, J., and Misra, C.: On-road Heavy-duty Vehicle Emissions Monitoring System, Environmental Science & Technology, 49, 1639-1645, 10.1021/es505534e, 2015.

Lefohn, A. S., Shadwick, D., and Oltmans, S. J.: Characterizing changes in surface ozone levels in metropolitan and rural areas in the United States for 1980–2008 and 1994–2008, Atmospheric Environment, 44, 5199-5210, 2010.

Marr, L. C., and Harley, R. A.: Modeling the Effect of Weekday– Weekend Differences in Motor Vehicle Emissions on Photochemical Air Pollution in Central California, Environmental science & technology, 36, 4099-4106, 2002.

McDonald, B. C., Gentner, D. R., Goldstein, A. H., and Harley, R. A.: Long-term trends in motor vehicle emissions in US urban areas, Environmental science & technology, 47, 10022-10031, 2013.

McDuffie, E. E., Edwards, P. M., Gilman, J. B., Lerner, B. M., Dubé, W. P., Trainer, M., Wolfe, D. E., Angevine, W. M., deGouw, J., and Williams, E. J.: Influence of oil and gas emissions on summertime ozone in the Colorado Northern Front Range, Journal of Geophysical Research: Atmospheres, 121, 8712-8729, 2016.

Pétron, G., Frost, G., Miller, B. R., Hirsch, A. I., Montzka, S. A., Karion, A., Trainer, M., Sweeney, C., Andrews, A. E., Miller, L., Kofler, J., Bar-Ilan, A., Dlugokencky, E. J., Patrick, L., Moore, C. T., Ryerson, T. B., Siso, C., Kolodzey, W., Lang, P. M., Conway, T., Novelli, P., Masarie, K., Hall, B., Guenther, D., Kitzis, D., Miller, J., Welsh, D., Wolfe, D., Neff, W., and Tans, P.: Hydrocarbon emissions characterization in the Colorado Front Range: A pilot study, Journal of Geophysical Research: Atmospheres, 117, n/a-n/a, 10.1029/2011jd016360, 2012.

Pétron, G., Karion, A., Sweeney, C., Miller, B. R., Montzka, S. A., Frost, G. J., Trainer, M., Tans, P., Andrews, A., and Kofler, J.: A new look at methane and nonmethane hydrocarbon emissions from oil and natural gas operations in the Colorado Denver-Julesburg Basin, Journal of Geophysical Research: Atmospheres, 119, 6836-6852, 2014.

[revised manuscript text omitted]

Pétron, G., Karion, A., Sweeney, C., Miller, B. R., Montzka, S. A., Frost, G. J., Trainer, M., Tans, P., Andrews,
A., and Kofler, J.: A new look at methane and nonmethane hydrocarbon emissions from oil and natural
gas operations in the Colorado Denver-Julesburg Basin, Journal of Geophysical Research: Atmospheres,
119, 6836-6852, 2014.
Pfister, G., Parrish, D., Worden, H., Emmons, L., Edwards, D., Wiedinmyer, C., Diskin, G., Huey, G., Oltmans,
S., and Thouret, V.: Characterizing summertime chemical boundary conditions for airmasses entering the
US West Coast, Atmospheric Chemistry and Physics, 11, 1769-1790, 2011.
Pollack, I., Ryerson, T., Trainer, M., Parrish, D., Andrews, A., Atlas, E. L., Blake, D., Brown, S., Commane, R.,
and Daube, B.: Airborne and ground-based observations of a weekend effect in ozone, precursors, and
oxidation products in the California South Coast Air Basin, Journal of Geophysical Research: Atmospheres,
117, 2012.
Pusede, S., Gentner, D., Wooldridge, P., Browne, E., Rollins, A., Min, K.-E., Russell, A., Thomas, J., Zhang,
L., and Brune, W.: On the temperature dependence of organic reactivity, nitrogen oxides, ozone
production, and the impact of emission controls in San Joaquin Valley, California, Atmospheric Chemistry
and Physics, 14, 3373-3395, 2014.
Pusede, S. E., and Cohen, R. C.: On the observed response of ozone to $NO_x$ and VOC reactivity
reductions in San Joaquin Valley California 1995–present, Atmospheric Chemistry and Physics, 12, 8323-
8339, 10.5194/acp-12-8323-2012, 2012.
Reddy, P. J., and Pfister, G. G.: Meteorological factors contributing to the interannual variability of mid-
summer surface ozone in Colorado, Utah, and other western US states, Journal of Geophysical Research:
Atmospheres, 2016.
Singh, H. B., and Hanst, P. L.: Peroxyacetyl nitrate (PAN) in the unpolluted atmosphere: An important
reservoir for nitrogen oxides, Geophysical Research Letters, 8, 941-944, 1981.
Swarthout, R. F., Russo, R. S., Zhou, Y., Hart, A. H., and Sive, B. C.: Volatile organic compound distributions
during the NACHTT campaign at the Boulder Atmospheric Observatory: Influence of urban and natural gas
sources, Journal of Geophysical Research: Atmospheres, 118, 10,614-610,637, 10.1002/jgrd.50722, 2013.
Tai, A. P., Martin, M. V., and Heald, C. L.: Threat to future global food security from climate change and
ozone air pollution, Nature Climate Change, 4, 817-821, 2014.
Thompson, C. R., Hueber, J., and Helmig, D.: Influence of oil and gas emissions on ambient atmospheric
non-methane hydrocarbons in residential areas of Northeastern Colorado, Elementa: Science of the
Anthropocene, 2, 000035, 10.12952/journal.elementa.000035, 2014.
Thompson, M. L., Reynolds, J., Cox, L. H., Guttorp, P., and Sampson, P. D.: A review of statistical methods
for the meteorological adjustment of tropospheric ozone, Atmospheric environment, 35, 617-630, 2001.
Natural Gas - Data: https://www.eia.gov/, access: 4/15, 2017.

Warneke, C., Gouw, J. A., Edwards, P. M., Holloway, J. S., Gilman, J. B., Kuster, W. C., Graus, M., Atlas, E.,
Blake, D., and Gentner, D. R.: Photochemical aging of volatile organic compounds in the Los Angeles basin:
Weekday-weekend effect, Journal of Geophysical Research: Atmospheres, 118, 5018-5028, 2013.
Weiss-Penzias, P., Jaffe, D. A., Swartzendruber, P., Dennison, J. B., Chand, D., Hafner, W., and Prestbo, E.:
Observations of Asian air pollution in the free troposphere at Mount Bachelor Observatory during the
spring of 2004, Journal of Geophysical Research: Atmospheres, 111, 2006.
White, A., Darby, L., Senff, C., King, C., Banta, R., Koermer, J., Wilczak, J., Neiman, P., Angevine, W., and
Talbot, R.: Comparing the impact of meteorological variability on surface ozone during the NEAQS (2002)
and ICARTT (2004) field campaigns, Journal of Geophysical Research: Atmospheres, 112, 2007.
Zhou, X., and Geerts, B.: The influence of soil moisture on the planetary boundary layer and on cumulus
convection over an isolated mountain. Part I: observations, Monthly Weather Review, 141, 1061-1078,
2013.

| Site | Latitude | Longitude | Elevation (m) | Measurements |
|------|----------|-----------|---------------|--------------|
| CAMP | 39.7512 | -104.988 | 1591 | $O_3$ & $NO_2^*$ |
| Welby | 39.8382 | -104.955 | 1554 | $O_3$ & $NO_2^*$ |
| Carriage | 39.7518 | -105.031 | 1619 | $O_3$ |
| Fort Collins | 40.5775 | -105.079 | 1523 | $O_3$ |
| Greeley | 40.3864 | -104.737 | 1476 | $O_3$ |
| Rocky Flats | 39.9128 | -105.189 | 1784 | $O_3$ |
| I-25 | 39.7321 | -105.015 | 1586 | $NO_2^*$ |
| La Casa | 39.7795 | -105.005 | 1601 | $O_3$ & $NO_2^*$ |

Table 1. Summary of Measurements sites used in this analysis. Note that $NO_2^*$ refers to the $NO_2$ detected by the EPA reference method, and thus includes a fraction of $NO_y$ species.